# SPIN-Bench: How Well Do LLMs Plan Strategically and Reason Socially?

**Jianzhu Yao**[1],[*] **Kevin Wang**[2],[*] **Ryan Hsieh**[2], **Haisu Zhou**[2], **Tianqing Zou**[2],
**Zerui Cheng**[1], **Zhangyang Wang**[2], **Pramod Viswanath**[1]
[1]Princeton University
[2]The University of Texas at Austin
{jy0246, pramodv}@princeton.edu, {kevinwang.1839, atlaswang}@utexas.edu

## Abstract

Reasoning and strategic behavior in *social interactions* is a hallmark of intelligence. This form of reasoning is significantly more sophisticated than isolated planning or reasoning tasks in static settings (e.g., math problem solving). In this paper, we present *Strategic Planning, Interaction, and Negotiation* (**SPIN-Bench**)[1], a new multi-domain evaluation designed to measure the intelligence of *strategic planning* and *social reasoning*. While many existing benchmarks focus on narrow planning or single-agent reasoning, SPIN-Bench combines planning domain definition language (PDDL) tasks, competitive board games, cooperative card games, and multi-agent negotiation scenarios in one unified framework. The framework includes both a benchmark as well as an arena to simulate and evaluate the variety of social settings to test reasoning and strategic behavior of AI agents. We formulate the benchmark SPIN-Bench by systematically varying action spaces, state complexity, and the number of interacting agents to simulate a variety of social settings where success depends on not only methodical and step-wise decision making, but also *conceptual inference* of other (adversarial or cooperative) participants. Our experiments reveal that while contemporary LLMs handle *basic fact retrieval* and *short-range planning* reasonably well, they encounter significant performance bottlenecks in tasks requiring *deep multi-hop reasoning* over large state spaces and *socially adept* coordination under uncertainty. We envision SPIN-Bench as a catalyst for future research on robust multi-agent planning, social reasoning, and human–AI teaming.

## 1 Introduction

Large Language Models (LLMs) have recently demonstrated remarkable proficiency in generating coherent text, contextual understanding, and a variety of agentic tasks (Schick et al., 2023; DataBassGit, 2023). This progress has opened numerous avenues for deploying LLM-based agents in real-world applications such as digital assistance, complex decision-making support, and collaborative human–AI systems (Liu et al., 2023; Mialon et al., 2024). Yet, many tasks in these domains demand more than simple question-answering or short-range inference: they hinge on *strategic planning*, where large action/state spaces and multi-step goal formulations require sophisticated *long-horizon* thinking (Sawada et al., 2024; Duan et al., 2024b).

Beyond step-wise decision-making, modern AI systems increasingly need *social intelligence*: the ability to negotiate, cooperate, and reason about other agents' hidden goals and beliefs (Hou et al., 2024; Cross et al., 2024). Environments such as *Diplomacy* (Meta Fundamental AI Research Diplomacy Team (FAIR) et al., 2022), *Avalon* (Light et al., 2023), *Werewolf* (Xu et al., 2023), and various negotiation-based games (Liang et al., 2023; Abdelnabi

---

[*]Equal contribution
[1]Project page: https://spinbench.github.io/. Code: https://github.com/spinbench/spinbench

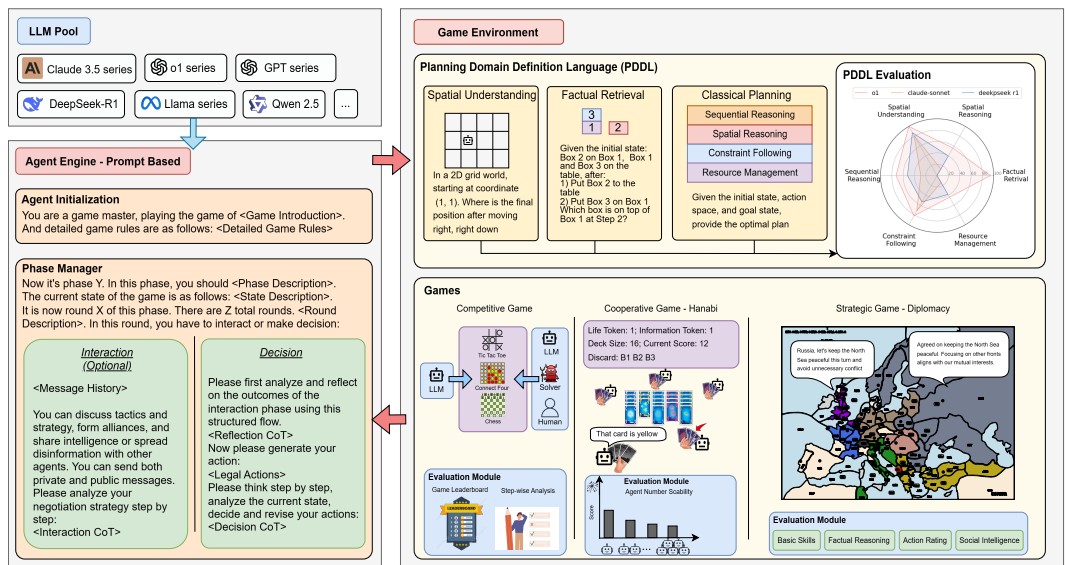

Figure 1: Overview of the *Strategic Planning, Interaction, and Negotiation* (**SPIN-Bench**) framework, highlighting its two core components: (1) the **Game Agent**, which encompasses the LLMs and their adaptive prompting, and (2) the **Environment** and **Evaluation** subsystem, which manage game logic, track interactions, and quantify performance.

et al., 2023; Hua et al., 2024) require not just logical strategies but also alliance formation, perspective-taking, and the capacity to handle incomplete or uncertain information.

Taken together, these abilities can be seen as both **methodical, step-wise planning** (executing sequences of actions in a possibly large search space) and **conceptual inference** (reasoning about hidden states, other players' intentions, and partial observability). Successful multi-agent collaboration or competition arises when an AI system integrates both these aspects, adapting its plan based on the changing behavior of other agents.

However, few existing benchmarks provide a comprehensive, unified setting to rigorously test how LLMs balance these two intertwined requirements—strategic long-horizon planning and multi-agent social reasoning. To address these gaps, we introduce *Strategic Planning, Interaction, and Negotiation* (**SPIN-Bench**), a new evaluation framework specifically designed to capture both:

1. *Strategic planning* across single-agent formal tasks (PDDL) and highly interactive board games with large action and state spaces.

2. *Social intelligence* in multi-agent scenarios requiring negotiation, cooperation, or competition, as exemplified by *Hanabi* and *Diplomacy*.

Crucially, SPIN-Bench integrates a rich array of tasks that systematically scale up the complexity of the environment, thus exposing where and how LLMs' planning or social reasoning falls short. We evaluate a broad spectrum of popular open-source and state-of-the-art closed-source models, revealing significant performance bottlenecks as tasks become more intricate and agent interactions intensify.

Our key findings are two fold. First, although models such as o1 exhibit competitive performance in certain long-horizon planning scenarios, they still falter in *deep multi-hop reasoning* once the action and state spaces expand substantially. Second, despite their advanced language generation capabilities, most current LLMs underperform in cooperative and negotiation tasks, suggesting a deficiency in *social intelligence* under complex strategic conditions. Intriguingly, we observe that large-scale social interaction can *negatively affect* the chain-of-thought coherence in otherwise strong planning models like o1—an interplay that has not been scrutinized in simpler benchmarks.

In summary, our work makes the following contributions:

1. **A Unified Evaluation:** We introduce a single framework that spans formal planning (PDDL), board and card games, and negotiation-based scenarios, bridging previously siloed areas of research.

2. **Systematic Strategic Reasoning Analysis:** We expose fundamental limitations in how LLMs handle increasing action/state complexity, thus identifying future directions for improving long-horizon planning.

3. **Social Intelligence Assessment:** By testing multi-agent negotiation, cooperation, and alliance-building, we highlight critical gaps between LLM performance and human baselines in strategic social contexts.

Altogether, SPIN-Bench underscores current insufficiencies of LLMs in *long-horizon, multi-agent* tasks. We hope that our benchmark will motivate deeper research into *strategic planning* and *social reasoning*, paving the way for more capable human–AI partnerships and progress toward broader AI goals.

## 2 Related work

**Evaluation of LLM in Planning**  LLMs show promise in methodical, step-wise planning but still face notable limitations. Early work revealed fundamental reasoning failures in PDDL-based tasks (Hao et al., 2023; Valmeekam et al., 2023; 2024a; Wang et al., 2024). Recent improvements in test-time scaling have enabled models to achieve over 95% accuracy on discrete PDDL benchmarks (Valmeekam et al., 2024b; Wang et al., 2024). However, these evaluations do not capture the deeper challenges of long-horizon planning. As outlined in Ghallab et al. (2004), planning involves understanding, reasoning, and execution; we further investigate at which stage each component begins to falter.

**LLMs for gaming and social intelligence**  Games offer an appealing testbed for LLMs because they often demand multi-step reasoning, strategic planning, and multi-agent interaction. Recent work on single-game evaluations has explored domains such as Minecraft (Gong et al., 2024), Avalon (Light et al., 2023), and Werewolf (Xu et al., 2023), but these narrow settings limit generalizability. Broader benchmarks incorporate multiple game types (Paglieri et al., 2024; Ruoss et al., 2024; Zhang et al., 2024; Duan et al., 2024b;a; Costarelli et al., 2024), and multi-agent frameworks emphasizing coordination or competition (Huang et al., 2024; Cross et al., 2024; Agashe et al., 2024), yet critical dimensions—like *open-ended negotiation*, evolving cooperation versus conflict, and richer social dynamics—remain underexplored. Existing work builds a social AI data infrastructure (Li et al., 2024), and raises several core technical challenges to advance social intelligence (Mathur et al., 2024), while other lines of research delve deeper into social intelligence in game-based contexts, investigating theory of mind (Cross et al., 2024; Hou et al., 2024; Sclar et al.), game-theoretic decision-making (Chen et al., 2023; Akata et al., 2023; Abdelnabi et al., 2023), and dialogue-driven challenges (Qiao et al., 2023; Chen et al., 2024; Duan et al., 2024b; Abdulhai et al., 2023; Zhou et al., 2024), but often within constrained strategies or limited forms of social dynamics. In contrast, our SPIN-Bench unifies strategic planning and social intelligence in a single testbed - combining formal PDDL analysis, multi-agent cooperation and competition, open-ended dialogues, and direct comparisons against optimal solvers and human baselines-to uncover richer insights into the interplay between *step-wise action selection* and *conceptual inference* in extended-horizon gaming environments.

## 3 SPIN-Bench: evaluating game-planning competency in LLMs

We now present SPIN-Bench, a comprehensive benchmark designed to evaluate how well LLMs integrate *methodical, step-wise planning* with *conceptual social reasoning* across a diverse range of tasks. As depicted in Figure 1, SPIN-Bench encompasses four major components—*PDDL Tasks*, *Competitive Games*, *Cooperative Games*, and *Strategic Games*—each reflecting increasingly intricate environments and forms of multi-agent interaction. In this

section, we first formalize the *problem settings* underlying these tasks (Section 3.1), then introduce the specific game environments (Section 3.2) and describe our *benchmark construction* and *evaluation metrics* (Sections 3.3–3.4).

### 3.1 Problem definition: three settings for action sequence generation

SPIN-Bench organizes automated action selection into three progressively complex frameworks — classical (single-agent, deterministic) planning, multi-agent (cooperative or competitive) games, and strategic games featuring dynamic alliances and negotiation. Each setting incrementally increases demands on an LLM's planning and social reasoning abilities. Formal definitions, task formulations, and illustrative examples for all three settings appear in Appendix B.1. By exploring these three categories in tandem, SPIN-Bench systematically illuminates how LLMs grow from simpler single-agent planners into more complex negotiators and alliance builders, testing both *mechanical planning* skills and *social inference* in dynamic, multi-turn environments.

### 3.2 Game taxonomy and environments

Figure 1 overviews how these *three settings* map onto specific SPIN-Bench tasks. In total, we integrate **PDDL-based planning**, **competitive board games**, **cooperative card games**, and **negotiation-focused strategic games**:

**PDDL.** Within SPIN-Bench, classical planning tasks offer a controlled environment to rigorously test core reasoning skills. We employ three subcategories: *(i) factual retrieval* in domains like `elevator`, `grid`, and `floortile` with progressively expanding state spaces; *(ii) spatial reasoning* in a modified `floortile` scenario that challenges models to track agent location from relative instructions; *(iii) full classical planning* tasks spread across 21 domains (1,280 tasks), each involving sequential decisions, resource management, and constraint handling. Appendix B.2 provides domain-specific details. By starting with simpler tasks and escalating to complex multi-step planning, SPIN-Bench pinpoints whether an LLM's failures arise from inadequate state tracking, partial-order reasoning, or chain-of-thought breakdowns.

**Competitive games.** To evaluate adversarial reasoning, we include three widely studied turn-based board games of escalating complexity: `Tic-tac-toe`, `Connect Four`, and `Chess`. Each underscores a key facet of competitive strategy: from short-range lookahead and forcing moves to deeper multi-step tactics. Game introduction is shown in Appendix B.3. These games also enable direct comparisons against established AI solvers (Minimax (von Neumann, 1928) or Stockfish (The Stockfish developers, 2025)) and against human-level baselines, providing a clear gauge of how far LLMs are from optimal or near-optimal play.

**Cooperative game (Hanabi).** For multi-agent collaboration under imperfect information, SPIN-Bench leverages the cooperative card game `Hanabi`. Here, each player observes others' cards but not their own, and must communicate implicitly to play cards in the correct order. This format demands not just straightforward planning but also trust-building, reasoning about other agents' beliefs and intentions, and coordinated actions — essential for more advanced cooperative planning. Previous work (Bard et al., 2020) demonstrated that Hanabi requires advanced theory of mind reasoning, making it well-suited for evaluating social intelligence in collaborative settings. More details about how to play Hanabi are shown in Appendix B.4.

**Strategic game (Diplomacy).** Finally, SPIN-Bench incorporates `Diplomacy` (Meta Fundamental AI Research Diplomacy Team (FAIR) et al., 2022), a complex multi-player board game in which negotiation, dynamic alliances, and strategic betrayal are integral. Players exchange messages to coordinate or mislead, while simultaneously issuing movement orders to capture territories. Game introduction and tutorial are shown in Appendix B.5. Because negotiation is central, LLMs are tested on both *long-horizon planning* and *social intelligence*—managing alliances, inferring hidden intentions, and adapting their strategies

in response to evolving alliances. This setting is especially suited for evaluating how chain-of-thought reasoning coexists with dynamic social interaction under partial cooperation and partial competition. It captures a broader spectrum of *realistic multi-agent behavior* that blends both cooperation and competition within the same session.

### 3.3 Benchmark construction

**Classical planning data generation.** SPIN-Bench-Classical is built by adapting established PDDL competition benchmarks (Seipp et al., 2022) and designing additional domains to target specific LLM capabilities (factual retrieval, spatial updates, etc.). Problem instances are generated via a systematic randomization pipeline and validated with Fast Downward (Helmert, 2006) or SMTPlan (Cashmore et al., 2016). The VAL system (Planning, 2019) ensures solution feasibility. This tiered approach captures how LLMs transition from simpler factual tasks to multi-step planning under growing constraints.

**LLM game agent interface.** We implement a unified interface that supplies each model with the current state, relevant history, and a list of legal actions, alongside game-specific Chain-of-Thought (Wei et al., 2022) prompts. In *Diplomacy*, we integrate the Diplomacy game engine (Paquette et al., 2020) and include both public and private messages. Models may retry illegal moves up to ten times before incurring an automatic loss. Full details on engine integration, prompt templates and domain rules, appear in Appendix B, E.

### 3.4 Evaluation metrics

**Rule-based metrics.** For classical planning tasks, we measure both *accuracy* and *N-Step Look Ahead* (see Appendix B.2 for the formula). In *competitive games*, we compare LLM moves against solver-generated top-$k$ actions and track head-to-head outcomes against the solver. We also maintain a leaderboard of *internal Elo* ratings (do not directly reflect performance among humans). This helps us gauge the "skill gap" among different LLMs and compare them against human baselines. For *Hanabi*, we use the final score to assess cooperative efficiency. Meanwhile, *Diplomacy* performance is evaluated through factual consistency, order-level success rates, and final game outcomes (see Appendix B.5 for details).

**LLM-assisted negotiation metrics.** In negotiation-heavy settings like *Diplomacy*, we use six fine-grained, *LLM-assisted* metrics: *(1)* alignment of messages with stated negotiation strategies, *(2)* proposal acceptance rate, *(3)* mutual vs. one-sided benefit of proposed deals, *(4)* peace vs. conflict inclination, *(5)* evidence of perspective-taking, and *(6)* conditional negotiation. We then prompt `Claude 3.7 Sonnet` to annotate chat logs along these dimensions, enabling a deeper look into each LLM's social intelligence, strategic coherence, and ability to balance persuasion with deception. Metric details appear in Appendix B.5.2. We also conduct human annotation experiments to verify the accuracy of LLM annotations in Appendix C.3.3.

## 4 Experiments on SPIN-Bench

In this section, we provide a comprehensive evaluation of how well Large Language Models (LLMs) handle the range of SPIN-Bench tasks introduced in Section 3. Specifically, we investigate whether current LLMs:

1. Manage *core planning competencies*, such as spatial reasoning, factual retrieval, and constraint following.
2. Cope with *rising action complexity* and larger multi-agent interactions, comparing their performance to both optimal solvers and human baselines.
3. Demonstrate *social intelligence* in negotiation and cooperative settings.

In what follows, we first outline the overall experimental setup (Section 4.1). Next, we detail results on core planning skills (Section 4.2), examine the impact of action complexity

and agent scaling (Section 4.3) and evaluate social intelligence and negotiation behaviors (Section 4.4). The reproducibility statement of all our evaluations is in Appendix A. Our trajectory viewer present game trajectories and examples in all benchmark settings.

## 4.1 Model and configurations

We evaluate a suite of popular LLMs that encompasses both closed-source commercial and open-source systems. The **closed-source models** include GPT-4o (Hurst et al., 2024), GPT-4o mini (OpenAI, 2024a), o1 (Jaech et al., 2024), o1-preview (OpenAI, 2024c), o1-mini (OpenAI, 2024b), o3-mini (OpenAI, 2025), o4-mini (OpenAI, 2025), GPT-4-turbo, GPT-3.5-turbo, Claude 3.5 Sonnet (Anthropic, 2024a), and Claude 3.5 Haiku (Anthropic, 2024b). Our **open-source models** include DeepSeek-R1 (Guo et al., 2025), Llama4 Maverick (MetaAI, 2025), Llama3-70b, Llama3.1-70b, Llama3.2-3b, Llama3.3-70b (MetaAI, 2024a; AI, 2024; MetaAI, 2024b;c), Qwen2.5-72b (Team, 2024), and Mistral-7b (Jiang et al., 2023). Since Diplomacy exhibits the largest branching factor in our benchmark, we first conducted a basic skill evaluation to identify which models should advance to the full Diplomacy task. Detailed results for the basic skill evaluation are provided in Appendix C.3.1. We excluded Diplomacy from the benchmark average score in Table 1 as few models play it well.

| | Classical Planning | | Competitive Games | | | | | Collaborative: Hanabi | | | | Avg. |
|---|---|---|---|---|---|---|---|---|---|---|---|---|
| Model | Plan Acc ↑ | N-Step↑ | $TTT_{DR}$↑ | $C4_{DR}$↑ | $CH_{DR}$↑ | $C4_{T3}$↑ | $CH_{T3}$↑ | 2P↑ | 3P↑ | 4P↑ | 5P↑ | Score↑ |
| o1 | **58.59** | **16.09** | 70.0 | 0.0 | 0.0 | 83.1 | 45.9 | **16.4** | 14.8 | **14.8** | 14.2 | **49.8** |
| o4-mini | 46.79 | 11.52 | **80.0** | 0.0 | 0.0 | 81.1 | 50.5 | 12.8 | 11.0 | 12.6 | 13.2 | 45.6 |
| o1-mini | 13.20 | 1.95 | 50.0 | 0.0 | 0.0 | **87.0** | 36.5 | 6.8 | 7.4 | 11.4 | 10.2 | 33.0 |
| o3-mini | 51.25 | 13.04 | 20.0 | 0.0 | 0.0 | 74.2 | **52.8** | 8.8 | 7.6 | 8.8 | 8.0 | 33.1 |
| GPT-4o | 8.75 | 0.60 | 0.0 | 0.0 | 0.0 | 84.1 | 32.2 | 6.6 | 4.8 | 4.8 | 4.6 | 20.8 |
| GPT-4-turbo | 5.62 | 0.13 | 60.0 | 0.0 | 0.0 | 83.8 | 38.7 | 5.2 | 5.6 | 5.0 | 6.0 | 27.5 |
| Claude 3.5 Sonnet | 20.55 | 4.44 | 60.0 | 0.0 | 0.0 | 78.9 | 49.5 | 8.2 | 9.4 | 7.4 | 8.4 | 34.3 |
| Claude 3.5 Haiku | 4.22 | 0.30 | 50.0 | 0.0 | 0.0 | 69.6 | 35.9 | 2.4 | 4.0 | 2.8 | 2.8 | 20.8 |
| DeepSeek R1 | 44.30 | 10.71 | 10.0 | 0.0 | 0.0 | 78.9 | 47.8 | 6.0 | **16.0** | 11.3 | 13.0 | 36.6 |
| Llama4 Maverick | 13.05 | 1.69 | 0.0 | 0.0 | 0.0 | 75.4 | 51.3 | 3.0 | 4.2 | 4.4 | 5.6 | 20.8 |
| Llama-3.3-70b | 5.78 | 0.32 | 0.0 | 0.0 | 0.0 | 79.5 | 25.4 | 2.4 | 0.8 | 0.6 | 1.4 | 13.1 |

Table 1: **Results for Classical Planning, Competitive Games, and Collaborative Game (Hanabi).** TTT, C4, CH: Tic Tac Toe, Connect Four and Chess. DR shows LLMs' draw rate (%) against solvers(for chess, we use stockfish-20). T3 shows the top-3 moves percentages among all games against the solver. Hanabi reports average score for player counts 2–5. Avg Score is calculated as the average of Plan Acc and all game metrics, with Hanabi scores normalized to percentages (divided by full score 25).

## 4.2 Core planning competencies: factual retrieval, spatial reasoning and constraint following

We first examine *why* LLMs might falter on long-horizon planning by isolating two crucial facets: (*i*) factual retrieval and (*ii*) spatial reasoning. In addition, we perform an error analysis that highlights common failure modes.

**Factual retrieval.** In Diplomacy, we design and categorize several factual queries into *one-hop* (e.g., "Which locations does Russia control?") vs. *multi-hop* (e.g., "Which territories can France attack, and how many units of support are needed at least?") to further check models' factual retrieval in a highly strategic environment. As summarized in Figure 2, nearly all LLMs do well on basic location or adjacency checks but degrade by a large margin on "Attackable" and "Attack Analysis," which demand deeper, multi-hop inference. Again, o1 and o1-preview lead but still exhibit significant drops com-

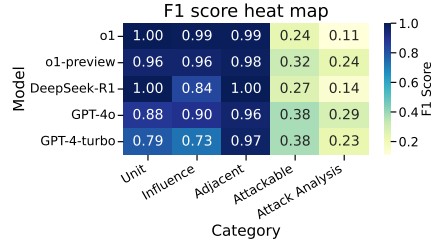

Figure 2: Heatmap displaying the F1 scores across evaluation categories in *Diplomacy*.

pared to simpler tasks. We also examined the factual retrieval performance of models within the classical planning task in Appendix C.1. Our evaluation indicates that recent reasoning models can achieve accuracy exceeding 90%, while other models exhibit significant performance degradation as problem complexity increases.

**Spatial reasoning.** To investigate whether LLMs' planning deficits stem from weaker spatial understanding, we designed tasks based on PDDL format that require each model to track positions across sequences of relative movements. In these tasks, the LLM is given an initial spatial coordinate (e.g., (2,2)) along with a series of movements(up, down, left, right), and is tasked with computing the final coordinate. We evaluated all models on 90 scenes, each with a varying number of steps, to assess their spatial reasoning capabilities. Figure 3(c) plots the accuracy of each model against the length of the movement trajectory. Notably, o1-mini and GPT-4o exhibit declining performance as the number of steps increases, whereas o1 sustains *perfect* accuracy (100%) up to 29 steps.

**Error analysis.** In **classical planning**, we categorized the errors into three distinct types: (1) *Breaking Constraints (BC)* includes execution failures, type mismatches, and structural issues, high BC reflecting difficulty in executing the reasoning; (2) *Goal Not Satisfied (GS)* refers to plans that satisfy all constraints yet fail to achieve the goal state; and (3) *Others* encompasses parsing errors, format inconsistencies, and miscellaneous validation issues. The results reveal a striking contrast: while Claude 3.5 Sonnet achieves a relatively low BC error (28.59%), it suffers from an exceptionally high GS error (44.77%). In contrast, DeepSeek R1—employing reinforcement learning–based training—drastically cuts the GS error to just 3.65%. This highlight that RL-based training LLM is particularly effective at reasoning, but failed in execution. The detailed quantitative breakdown is provided in Table 2.

In **competitive board games**, illegal-move rates scale sharply with complexity: minimal in Tic-Tac-Toe, moderate in Connect Four, and highest in Chess (Appendix C.2), underscoring that rule compliance becomes a critical bottleneck as branching factors grow. In **Diplomacy**, o1 achieves the highest basic order success rate—20–30% above other models—but all models' performance deteriorates on multi-step and multi-agent actions (e.g., support moves), revealing new failure modes rooted in deeper strategic and alliance reasoning (Table 3).

| Model | BC (%)↓ | GS (%)↓ | Other (%)↓ |
|---|---|---|---|
| o1 | **17.97** | 17.89 | **5.55** |
| o1-mini | 68.91 | 5.94 | 11.95 |
| GPT-4o | 81.56 | **3.59** | 6.09 |
| Claude 3.5 Sonnet | 28.59 | 44.77 | 6.09 |
| Claude 3.5 Haiku | 72.03 | 15.23 | 1.76 |
| DeepSeek R1 | 48.28 | 3.65 | 1.87 |

Table 2: Error breakdown by model for the total of 1,280 problems.

| Model | 4 Agents (w/o vs. w. Negotiation) | | | | | | Negotiation Metrics: Social Intelligence | | | | | |
|---|---|---|---|---|---|---|---|---|---|---|---|---|
| | Move↑ | Attack↑ | SS↑ | SO↑ | SC↑ | CR↑ | AlignR↑ | AcceptR↑ | Mut:1S | P:C | Persp.↑ | Cond.↑ |
| o1 | 0.70/0.85 | 0.44/**0.83** | 0.45/0.46 | 0.00/0.00 | **17**/10 | 37/18 | **1.000** | **0.673** | 100.0 | 9.26 | 0.33 | 0.36 |
| GPT-4o | 0.55/0.62 | 0.33/0.37 | 0.30/0.29 | **0.25**/0.40 | 15/**17** | 28/**29** | 0.994 | 0.373 | 36.6 | 3.16 | 0.33 | 0.04 |
| GPT-4-turbo | 0.47/0.66 | 0.14/0.36 | 0.23/0.30 | 0.00/0.43 | 7/8 | 14/19 | 0.995 | 0.437 | 166.0 | 2.76 | 0.45 | 0.03 |
| DeepSeek-R1 | 0.64/0.65 | 0.47/0.47 | 0.35/0.43 | 0.00/**0.67** | 16/16 | 22/27 | 0.982 | 0.416 | 5.6 | 0.95 | 0.21 | **0.51** |
| o1-preview | **0.82**/**0.87** | **0.69**/0.67 | **0.46**/0.40 | 0.00/0.17 | 15/11 | **38**/26 | 0.994 | 0.479 | 27.3 | 2.04 | 0.44 | 0.15 |
| Claude 3.5 Haiku | 0.48/0.37 | 0.23/0.20 | 0.38/0.07 | 0.00/0.00 | 0/5 | 1/7 | 0.996 | 0.409 | 43.0 | 0.83 | **0.47** | 0.07 |

Table 3: **Diplomacy results for the 4-agent setting**, showing (without negotiation / with negotiation (x/y)) success rates for 4 types of orders, as well as the number of Supply Centers (SC) and Controlled Regions (CR) when game finished. More results for multi-agent settings are shown in Table 10. On the right side are negotiation metrics from left to right: Alignment Ratio, Acceptance Rate, Mutual:1Side, Peace:Conflict, Perspective, and Conditional Negotiation.

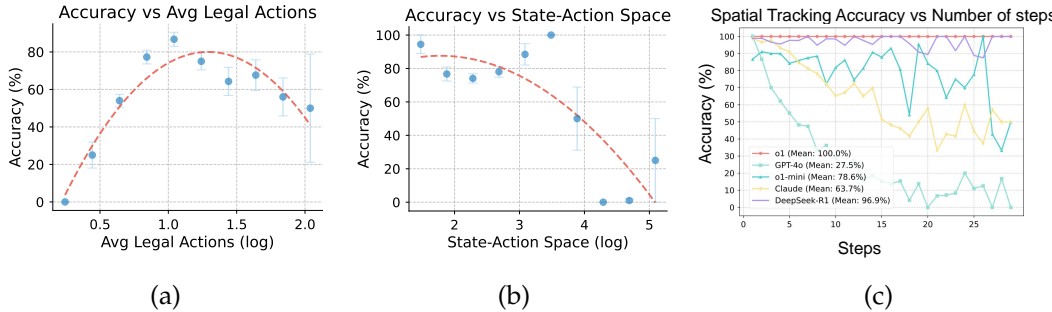

Figure 3: Relationship between accuracy and action space complexity. (a) Accuracy versus average number of legal actions for o1 model. (b) Accuracy versus grounded actions for o1 model. (c) Spatial reasoning accuracy across different models, where Claude refers to Claude 3.5 Sonnet.

### 4.3 Impact of action complexity and multi-agent scale on planning performance

#### 4.3.1 Complex or expansive action spaces

**PDDL domains.** We further examine the results of 10 domains in classical planning tasks by analyzing two key measures: the average number of legal actions available at each step and the total state–action space, defined as the complete set of possible state–action pairs given an initial state and action space. To evaluate their impact on model performance, we calculate the correlation between accuracy and these two metrics. As illustrated in Figure 3(a) and (b), accuracy demonstrates a stronger negative correlation with the total state–action space compared to the average number of legal actions. This implies that cognitively, the model "carries" the burden of large future branching factors, even if only a few choices are valid at each step.

**Competitive board games.** Across all three games, solver-level engines **never lose**, and LLMs fall short of optimal play (Table 1; full results in Appendix 8). In Tic-Tac-Toe, advanced models (e.g., o1, GPT-4-turbo, Claude 3.5 Sonnet) occasionally force a draw, but even here they underperform the perfect solver. In Connect Four and Chess, every LLM lost 100% of matches against solver-level engines. Top k action percentage analysis in Appendix(Figure 7) shows that while LLMs sometimes pick optimal moves in Connect Four, their accuracy drops drastically in Chess, underscoring how deeper tactics and branching expansions are beyond current LLMs' capacity. Internal Elo ratings(Table 4) rank o1-preview and o1-mini highest, demonstrating strongest strategic planning capabilities, further indicating the advantages of test time scaling in strategic games. Additionally, Claude 3.5 Sonnet and GPT-4o follows, indicating a superior ability to handle complex strategies compared to their counterparts, yet all remain faw below solver or human-expert performance. These results highlight a significant gap in strategic reasoning and search depth, underscoring current LLM limitations in planning over large action spaces.

| Model | TTT | C4 | CH |
|---|---|---|---|
| *Human* | 1415 | 1692 | - |
| o1-preview | 1263 | 1377 | 1395 |
| o1-mini | 1205 | 1083 | 1247 |
| Claude 3.5 Sonnet | 1138 | 942 | 1196 |
| GPT-4o | 1025 | 980 | 1255 |
| Claude 3.5 Haiku | 983 | 907 | 1041 |
| Qwen2.5:72b | 944 | 877 | 1150 |
| Llama3.3 | 944 | 878 | 1146 |
| Llama3.1:70b | 936 | 829 | 1164 |
| GPT-4o-mini | 912 | 925 | 842 |
| GPT-4-turbo | 857 | 1007 | 1265 |
| Llama3:70b | 856 | 993 | 1155 |
| Mistral:7b | 855 | 800 | 243 |
| Llama3.2:3b | 836 | 821 | 421 |
| GPT-3.5-turbo | 829 | 890 | 480 |

Table 4: Internal Elo Ratings of LLMs (Initial Elo: 1000)

#### 4.3.2 Scaling number of agents

In **Hanabi**, we vary the number of players from 2 to 5 and track final scores. Table 1 indicates that o1 remains the top performer, but its average score still declines from 16.4 (2-player) to 14.2 (5-player). This dip suggests that coordinating beliefs and actions among more agents still strains strong LLMs' capacity to maintain consistent strategies over multiple

incomplete information channels. However, models such as `o1-mini` and `DeepSeek-R1` display erratic performance. Their high variance and relatively low average scores suggest insufficient strategic reasoning capabilities for effective Hanabi play. In **Diplomacy**, we vary the agent number from 2 to 5. Detailed results of more multi-agent settings are shown in Appendix C.3.2. As the agent count grows (beyond 2-3 test seats for LLMs), we observe decreasing order accuracy, fewer successful attacks, and minimal supply-center gains once multi-party negotiations and hidden alliances come into play. Ultimately, LLMs lose traction in highly interactive scenarios, underscoring how partial observability and shifting alliances further intensify the multi-agent complexity.

## 4.4 Social intelligence and negotiation performance

While the preceding sections highlight LLMs' challenges in purely *strategic* or *planning-centric* domains, complex multi-agent settings further demand **social intelligence**—the ability to navigate cooperative, adversarial, and negotiated alliances. In this part of our evaluation, we compare LLM performance to *human baselines* and introduce explicit *negotiation phases* to assess whether social interactions degrade or enhance model behavior.

### 4.4.1 Cooperative scenarios vs. human baselines in Hanabi

We collected 54,977 human-played Hanabi games from BoardGameGeek, spanning 2- to 5-player settings. Figure 4 plots the human score distribution, highlighting quartiles (Q1–Q4) around a typical range of 15–25 points. While some LLMs do show patterns of declining performance with more agents (cf. Table 1), none approach even the first quartile of human scores. This underscores the *significant gap* in cooperative planning under hidden-information constraints—despite Hanabi's narrower branching factor relative to some competitive games. The results suggest a pervasive deficiency in *social intelligence* when LLMs must coordinate multiple incomplete information channels and track teammates' evolving knowledge states.

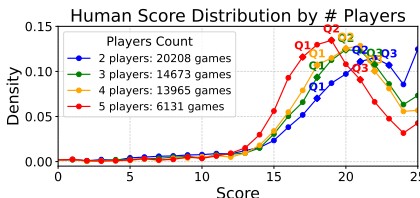

Figure 4: Hanabi score distribution by player count (54,977 games)

### 4.4.2 Negotiation in Diplomacy

**Overall impact on strategic play.** Negotiation phases in `Diplomacy`, which theoretically enable players to form or dissolve alliances, often produce counterintuitive effects on strong planners like `o1`, `o1-preview`. As shown in Table 3 and Table 10, `o1`'s all kinds of order numbers drop when negotiation is introduced, whereas other models sometimes stay the same or even improve slightly. And final game scores *Supply Centers* and *Controlled Regions* for `o1`, `o1-preview` also dropped by a large margin when enabling the negotiation, compared to other models. This result suggests that *intense social interaction* can disrupt planning coherence in otherwise capable LLMs, pointing to a tension between *extended chain-of-thought reasoning* and the cognitive overhead of real-time alliance-building, deception, or persuasion. In `DeepSeek-R1`, enabling negotiation features did not result in a significant performance decline. We attribute this to the model's inherent tendency to propose one-sided solutions and provoke conflict during interactions (see the next section).

**Negotiation message analysis.** We probe each model's negotiation style in greater depth by prompting `Claude 3.7 Sonnet` to annotate 1,864 messages across six-agent Diplomacy settings using six *LLM-assisted* metrics (see Appendix B.5.2 for details of the metrics; see Appendix C.3.3 for our human validation experiment of LLM annotation.). The results (see Table 3) enable the following key observations:

1. *High Strategy Alignment:* All models display alignment ratios above 0.98, implying that their negotiation messages largely match stated objectives.

2. *Persuasiveness:* o1 exhibits the highest acceptance rate ($\approx 0.67$), whereas GPT-4o is less effective ($\approx 0.37$).

3. *Mutual vs. One-Sided Benefits:* Models mostly propose mutually beneficial deals; GPT-4-turbo in particular tends not to push overtly one-sided offers.

4. *Peace vs. Conflict Messaging:* DeepSeek-R1 and Claude 3.5 Haiku demonstrate a relatively low peace ratio (0.95 and 0.83), showing their tendency to provoke conflicts, whereas o1 strongly prefers peaceful approaches (9.26).

5. *Perspective-Taking and Conditional Plans:* Although moderate across the board, Claude 3.5 Haiku leads in referencing other agents' intentions ($\approx 0.47$), while DeepSeek-R1 uses conditional tactics most frequently ($\approx 0.51$).

Despite these varied negotiation styles, most LLMs—o1 included—lack *flexibility* and *adaptivity* akin to skilled human negotiators. Rather than shifting strategies in response to betrayals or emergent alliances, models often cling to simplistic patterns of communication, or in o1's case, become overwhelmed by large volumes of social exchange. These observations reinforce that contemporary LLMs exhibit only nascent social intelligence in scenarios where partial cooperation and deception can dramatically alter long-horizon outcomes.

## 5 Limitations.

A primary limitation of this work lies in the use of *pre-defined prompt templates*. While we conducted detailed ablation studies on prompt design effects in our most complex environment Diplomacy (Appendix E.3), the static nature of these templates may not fully elicit models' latent social reasoning capabilities. Additionally, despite SPIN-Bench's breadth across multiple game-theoretic environments, it cannot exhaustively capture the complete spectrum of real-world strategic and social interactions. Our analytical scope deliberately focuses on two core dimensions of social intelligence (Goleman, 2006): Theory of Mind, which corresponds to social awareness, and negotiation, which exemplifies social facility. This targeted approach, while enabling deep analysis of these critical components, necessarily excludes other fundamental aspects of social intelligence, including primal empathy, social attunement, persuasion dynamics, and self-presentation strategies. Consequently, the generalizability of our findings to broader social reasoning contexts remains bounded. Future work should explore adaptive prompt generation techniques that dynamically adjust to model capabilities, expand SPIN-Bench to encompass additional social intelligence facets, and investigate longitudinal multi-round interactions that better approximate the temporal dynamics of real-world social intelligence.

## 6 Conclusion

In this paper, we introduced SPIN-Bench, a comprehensive benchmark designed to assess *strategic planning* and *social intelligence* in Large Language Models (LLMs) across multiple game environments of escalating complexity. Our experiments encompassed formal PDDL-based planning, competitive board games, cooperative incomplete-information scenarios, and negotiation-intensive strategic settings. By systematically varying action and state-space size, interaction modalities, and the number of agents, SPIN-Bench exposes critical limitations in today's LLMs. Despite recent improvements—particularly in short-range planning and factual recall—our findings reveal that most LLMs struggle with *long-horizon* tasks involving large branching factors and intricate multi-agent coordination. We also observe that *negotiation* and *social interplay* often degrade an LLM's chain-of-thought coherence, suggesting a tension between pure strategic reasoning and the cognitive overhead of dynamic alliances and potential deception. Overall, our work highlights pressing gaps in both *deep, multi-hop reasoning* and *robust social interaction*, pointing toward the need for novel architectural innovations, integrated planning modules, and more advanced training methodologies. We hope SPIN-Bench will serve as a foundation and catalyst for continued progress in developing LLMs that are more *strategically competent*, *socially adept*.

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

Within this supplementary material, we elaborate on the following points:

- Appendix A: Reproducibility statement
- Appendix B: Experiment settings for each game
- Appendix C: Supplementary experiments
- Appendix D: Prompt design and interaction flow

## A Reproducibility statement

We include a comprehensive setup and model specification in our code—scheduled for future release—to enable precise reproduction of our experimental results. But we also provide simple programs to run Tic Tac Toe, Connect Four, Chess, Hanabi and PDDL in our supplementary materials. Our package features scripts for running game engines, game solvers, the agent framework, and the entire benchmark arena. For commercial LLMs (e.g., Claude models), users can supply their own API keys. Additionally, to integrate new models into SPIN-Bench, we provide a generated PDDL dataset, solver scripts for three competitive games, multi-agent Diplomacy experiment settings, and all necessary interaction scripts.

**Compute.** We conducted all evaluations on an Ubuntu 22.04 Linux machine equipped with 64 cores, 1024 GB of RAM, and 8 NVIDIA RTX 6000 GPUs (each with 50GB VRAM). For LLMs accessible only via endpoints—such as OpenAI and Claude—we utilized their commercial endpoints, thereby eliminating the need for local GPU resources. Additionally, we performed a detailed analysis of token usage across the various benchmark tasks. All results will be released alongside our code.

## B Experiment settings

### B.1 Problem definition: three settings for action sequence generation

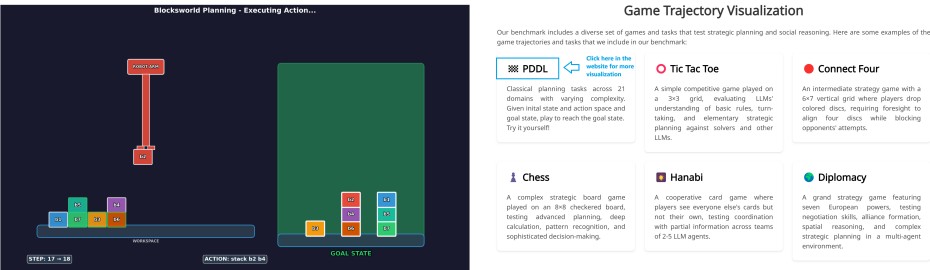

Figure 5: Sample visualization from our PDDL visualizer.

**To aid rule understanding, we've included animations for PDDL and interactive games. Try them here::** https://spinbench.github.io/tools/pddl/tra.html

**1. Classical Planning (Single-Agent, Deterministic).** A classical planning problem is typically defined by the tuple $\langle \mathcal{S}, s_{\text{init}}, \mathcal{S}_G, \mathcal{A}, f \rangle$, where:

- $\mathcal{S}$ is the set of all possible states of the environment,
- $s_{\text{init}} \in \mathcal{S}$ is the known initial state,
- $\mathcal{S}_G \subseteq \mathcal{S}$ denotes the goal region or goal states,
- $\mathcal{A}$ is the set of actions the agent can take,
- $f : \mathcal{S} \times \mathcal{A} \to \mathcal{S}$ is a deterministic transition function.

A valid plan is a finite sequence of actions $\pi = \langle a_1, a_2, \ldots, a_n \rangle$ that transforms $s_{\text{init}}$ into some goal state $s_g \in \mathcal{S}_G$. By design, this setting involves *one* decision-maker in a *fully observable*, *deterministic* environment, thereby providing the most controlled context for probing an LLM's fundamental step-wise planning capabilities.

**2. Multi-Agent Games (Cooperative or Competitive).** Multi-agent games generalize single-agent planning to multiple decision-makers $\{1, 2, \ldots, n\}$, each potentially with distinct or overlapping goals.

Formally, such a game can be written as

$$G = \langle \mathcal{S}, s_{\text{init}}, \{\mathcal{S}_{G_i}\}_{i=1}^n, \{\mathcal{A}_i\}_{i=1}^n, f \rangle,$$

where each agent $i$ has its own goal set $\mathcal{S}_{G_i}$ and action space $\mathcal{A}_i$. The transition function $f$ depends on the *joint* action of all agents. Games can be:

- *Competitive:* if $\mathcal{S}_{G_i} \cap \mathcal{S}_{G_j} = \varnothing$ for $i \neq j$, as in zero-sum formulations like Chess, requiring adversarial reasoning or minimax strategies.
- *Cooperative:* if all $\mathcal{S}_{G_i}$ coincide, thus encouraging agents to engage in joint planning under a shared objective (e.g., ).

**3. Strategic Games (Mixed Cooperation, Competition, and Negotiation).** Strategic games extend multi-agent scenarios by introducing *dynamic alliances, partial cooperation*, and *negotiation*. Here, the environment state may include past communication logs or partially hidden objectives, allowing agents to form or break alliances, re-evaluate shared goals, and even negotiate mid-game. This richer structure captures a broader spectrum of *realistic multi-agent behavior* that blends both cooperation and competition within the same session—*Diplomacy* (Meta Fundamental AI Research Diplomacy Team (FAIR) et al., 2022) being a prominent example.

## B.2 Classification and description of PDDL domains

Unlike existing LLM-based planning benchmarks, which typically focus on discrete-state PDDL problems, SPIN-Bench extends coverage to include numerical PDDL tasks with optimization goals. In these numeric domains, the objective is no longer simply to reach a discrete state but to achieve a specified numeric threshold or maximize/minimize a particular quantity. For example, in the *markettrader* domain, the goal is to accrue at least $100. LLMs perform worse on these numeric tasks than on standard discrete-state PDDL tasks, likely due to the combinatorial explosion and broader goal-state space inherent to numeric domains. A selection of the PDDL tasks is provided in the supplementary materials.

- **Spatial Domains**
    - *drone*: Focuses on controlling a drone across a map, planning flight paths and possibly collecting items or information.
    - *floortile*: Involves moving an agent over a grid of tiles to mark or paint them, ensuring proper coverage or pattern completion.
    - *grid*: Classic grid-navigation domain where an agent moves between cells to reach designated locations or achieve tasks.
    - *depots* (also resource management): Combines transportation and hoist operations to move crates between trucks and warehouses, requiring both navigation and efficient resource allocation.
    - *logistics*: Centers on delivering packages between locations using trucks and airplanes, emphasizing route planning and scheduling.
    - *rovers* (also resource management): Involves planetary rovers navigating terrain, collecting samples, and managing limited resources like fuel or battery power.
    - *sokoban*: Puzzle-based domain where an agent pushes crates in a warehouse grid, aiming to position them in specific target cells.

- *termes*: Models autonomous robots constructing structures in a grid by navigating, carrying blocks, and cooperating on building tasks.

- **Sequential Domains**
  - *assembly*: Entails sequentially combining parts to form a final product, highlighting the order of assembly steps.
  - *blocksworld*: Iconic stacking puzzle where blocks must be ordered and stacked under tight move constraints.
  - *briefcaseworld*: Tasks an agent with moving objects via a briefcase between locations, planning the order of loading/unloading.
  - *Multi_Agent_coordination*: Focuses on collaboration among multiple gripper agents, they need to cooperate to achieve the goal.
  - *cooperate_sequential_gripper* (also resource management): Multiple agents with limited grippers must coordinate object pickups, balancing the sequence of actions with shared resources.
  - *elevator* (also spatial): Handles transporting passengers or goods through building floors, requiring efficient elevator movements and scheduling.
  - *barman*: Bartender scenario where limited glasses and ingredients must be managed to prepare ordered drinks.

- **Resource Management Domains**
  - *counters*: Involves manipulating numerical counters under constraints to reach target values or states.
  - *markettrader*: A financial trading scenario requiring buying, selling, and resource (capital) management to achieve profit goals.
  - *satellite*: Space-based operations where satellites manage instruments, energy, and data storage to fulfill observation tasks.
  - *freecell*: Card puzzle variant with limited "free cells," each used as a temporary resource to reshuffle the card configuration.
  - *rovers* (also spatial): Rovers must navigate and handle tasks while managing finite consumables such as fuel or batteries.
  - *settlersnumeric*: Inspired by resource collection and expansion strategies, requiring numeric resource management to meet specified targets.
  - *sugar*: Involves transforming sugar among various states (solid, dissolved, etc.), requiring careful handling of processes and resource constraints.

**Metric: N Step Look Ahead**

$$\text{N Step Look Ahead} = \frac{\sum_{i=1}^{N}(C_i \times S_i)}{N},$$

where $N$ is the total number of tasks, $C_i$ indicates correctness (0 or 1), and $S_i$ is the required number of steps for task $i$. This weighted measure highlights performance gaps on more complex problems.

**B.3  Two-player competitive games**

| Game | Game Tree Complexity (as log to base 10) | Branching Factor (as log to base 10) | State Space Complexity (as log to base 10) |
|---|---|---|---|
| Tic Tac Toe | 5 | 4 | 3 |
| Connect Four | 21 | 4 | 13 |
| Chess | 123 | 35 | 44 |

Table 5: Game complexity statistics. Source: Wikipedia

**Game Introduction**  `Tic Tac Toe` is a classic and simple two-player game played on a 3x3 grid. Each player takes turns marking a space on the grid, with one player using "X" and the

other "O." The objective is to be the first to align three of your marks horizontally, vertically, or diagonally. Despite its straightforward rules, the game challenges players to think strategically, anticipate their opponent's moves, and block potential winning combinations. `Tic Tac Toe` is widely popular because it is easy to learn, quick to play, and requires minimal setup. So in our paper, we use `Tic Tac Toe` as the simplest setting testing LLMs' ability to understand basic rules and strategy.

`Connect Four` is a two-player strategy game played on a vertical grid with six rows and seven columns. Players take turns dropping colored discs into the grid, with the goal of being the first to align four of their discs horizontally, vertically, or diagonally. The game combines elements of planning and foresight, as players must both create winning opportunities and block their opponent's moves. Connect Four is known for its simple rules and engaging game-play. It is a little harder than `Tic Tac Toe`, so we include it to evaluate LLMs' capacity for intermediate-level strategic reasoning.

`Chess` is a timeless two-player strategy game played on an 8x8 checkered board. Each player controls an army of 16 pieces, including a king, queen, rooks, bishops, knights, and pawns, each with unique movements and abilities. The objective is to checkmate the opponent's king, putting it in a position where it cannot escape capture. `Chess` challenges players to think several moves ahead, employing tactics, strategy, and creativity. Revered as a game of intellect, it offers endless possibilities and rich complexities, making it an ideal test for LLMs' advanced planning and decision-making skills.

**Game Setting**

We have two settings for the two player competitive games, one is LLM-vs-Solver, and the other is LLM-vs-LLM.

For the **LLM-vs-Solver** setting, we make LLMs play against the most powerful game engines. For `Tic Tac Toe`, we implemented a solver using the Minimax algorithm to determine optimal moves by recursively evaluating all possible game states. It first checks if the game has ended in a win or draw. If not, it explores all legal moves, simulating each outcome by alternating between maximizing 'X''s score and minimizing 'O''s score. The best move is randomly chosen from the moves with highest values. This approach guarantees perfect play, and the solver's winning rate can be used as a strong benchmark for evaluating LLMs. For `Connect Four`, we use the Connect 4 solverPons (2023) implementation. For `Chess`, we use the well-known Stockfish engine, with different skill levels: 0, 5, 10, 15, 20 to compete with LLMs. Although chess engine Stockfish has external Elo ratings anchored to the CCLR Blitz Elos, since all LLMs are losing every game with even the lowest level of Stockfish, we didn't report the external elo rating for chess.

For the **LLM-vs-LLM** setting, we investigate different LLMs' performance in a competitive game setting. For each game, we tested 14 models in total, and every two models forming a pair, while introducing 91 game pairs. For each pair, we conduct 10 repetitive games for `Tic Tac Toe` and `Connect Four`, and 4 repetitive games for `Chess`. Players switch their positions after half of the pair competition. To provide a baseline on how average or expert humans perform against LLMs, especially in simpler games like `Tic Tac Toe` and `Connect Four`, we also include human performance in the leaderboard, with multiple game rounds between different LLMs and human to update the Elo rating.

In the leaderboard, we report *internal Elo* ratings (relative Elo between members of the population) for `Tic Tac Toe`, `Connect Four` and `Chess`. Elo rating is a well-established method widely used in chess and other competitive games, updated incrementally based on match outcomes and opponents' ratings. This metric effectively captures the skill gap between different LLMs playing these games, providing a quantitative measure of their relative performance.

For the step-wise evaluation metrics, we use the scores given by the solver and Stockfish, among all the trajectories in **LLM-vs-Solver** setting. For `Tic Tac Toe` and `Connect Four`, use the solver to generate scores for each move of LLM at each state. For `Chess`, we use Stockfish level 20 as the rating machine for step-wise analysis, on each step in the game between

LLMs and Stockfish level 0. We use "centipawns" as a measurement of the advantage of a chess board. A centipawn is equal to $1/100$ of a pawn. These values are essentials in computer chess to evaluate positions. For each pair of the game, we conduct 4 repetitive experiments.

### B.4 Multi-player cooperative game - Hanabi

**Game Introduction** (**Note:** The following game introduction is excerpted from the Hanabi Wikipedia Page)

Hanabi is a cooperative card game where players work together to create a series of fireworks by playing cards in ascending numerical order starting from 1. Each player holds their cards facing outward so that all players can see everyone else's cards but not their own. The objective is to play cards in sequence (1 through 5) for each color without making mistakes. There are 5 different colors and each color has cards numbered 1 to 5. The game begins with 8 available information tokens and 3 life/fuse tokens.

Play proceeds around the table; each turn, a player must take one of the following actions:

1. **Give Information:** The player points out the cards of either a given number or a given color in the hand of another player (examples: "This card is your only red card," "These two cards are your only 3s"). The information given must be complete and correct. Giving information consumes one information token.

2. **Discard a Card:** The player chooses a card from their hand and adds it to the discard pile, then draws a card to replace it. The discarded card is out of the game and can no longer be played. Discarding a card replenishes one information token.

3. **Play a Card:** The player chooses a card from their hand and attempts to add it to the cards already played. This is successful if the card is a 1 in a color that has not yet been played, or if it is the next number sequentially in a color that has been played. Otherwise a life/fuse token is consumed and the misplayed card is discarded. Successfully playing a 5 of any color replenishes one information token. Whether the play was successful or not, the player draws a replacement card.

The game ends immediately when either all life/fuse tokens are used up, resulting in a game loss, or all 5s have been played successfully, leading to a game win. Otherwise, play continues until the deck runs out, and for one full round after that. At the end of the game, the values of the highest cards in each color are summed, resulting in a total score out of a possible 25 points.

**Experiment Setting** In our Hanabi experiment, we tested 9 LLMs in total: o1, o1-mini, o3-mini, GPT-4o, GPT-4-turbo, Claude 3.5 Sonnet, Claude 3.5 Haiku, DeepSeek R1, Llama-3.3-70b. For each model, we have four settings: allowing the player number to range from 2 to 5. For example, model o1 with a player number of 3 means there are 3 separate o1 models playing the game. And we use the final score when the game ends as the score of each game setting.

### B.5 Strategic game - Diplomacy

#### B.5.1 Game introduction

(**Note:** The following game introduction is excerpted from the Diplomacy (game) Wikipedia Page)

Diplomacy is a 7-player turn based game, where players can use negotiation and strategy to control the most supply centers on the map. The players can move their units to different locations on the map, and can support other players' units to help them succeed. The game is played on a map of Europe, divided into territories and sea zones. The players can issue orders to their units to move, support, hold, or convoy. The game ends when one player controls 18 supply centers. 34 of the land provinces are supply centers. Possession of these

supply centers allows the powers who control them to raise and maintain armies and fleets. As they are also a central part of the game's victory conditions, they are the focus of much of the game's activity.

Each player is given three (save for Russia, which has four) home supply centers. These spaces are the starting point for their owning power's initial forces. The players can then build new units at these home supply centers as they capture further supply centers. New units can only be built on a power's home supply centers. If a power loses all of its home supply centers it may continue to play; however, it may not build new units until it has recaptured at least one of its home supply centers.

In Diplomacy, there are two types of units: Armies and Fleets. An army can travel in land spaces and coastal land spaces, and a fleet can travel in sea spaces and coastal land spaces. All units in Diplomacy move only one space at a time and only one unit may occupy any space at any time. The exception to this rule comes in the form of a successful convoy, where a convoyed army may travel multiple spaces depending on the length of the chain created by the convoying fleets. A convoyed army must embark from a coastal land province and land at a coastal land province.

Diplomacy proceeds by seasons, beginning in the year 1901, with each year divided into two main seasons: the "Spring" and "Fall" (Autumn) moves. Each season is further divided into negotiation and movement phases, followed by "retreat" or "disband" adjustments and an end-of-the-year Winter phase of new builds or removals following the Fall adjustments.

**Negotiation Phase**   In the negotiation phase, players discuss tactics and strategy, form alliances, and share intelligence or spread disinformation. Negotiations may be made public or kept private. Players are not bound to anything they say or promise, and no agreements are enforceable. Communication and trust are highly important; players must forge alliances with others and observe their actions to evaluate their trustworthiness. At the same time, they must convince others of their own trustworthiness while making plans to turn against their allies when least expected. A well-timed betrayal can be just as profitable as an enduring, reliable alliance.

**Movement Phase**   After the negotiation period, players write secret orders for each unit; these orders are revealed and executed simultaneously. A unit can move from its location to an adjacent space, support an adjacent unit to hold an area in the event of an attack, support another unit to attack a space into which it could move itself, or hold defensively. In addition, fleets may transport armies from one coast space to another when in a chain called a "convoy". Only one unit may occupy each region. If multiple units are ordered to move to the same region, only the unit with the most support moves there. If two or more units have the same highest support, a standoff occurs and no units ordered to that region move. A unit ordered to give support that is attacked has those orders canceled and is forced to hold, except in the case that support is being given to a unit invading the region from which the attack originated (in which case the unit that had been ordered to give support must retreat from, rather than hold, its position). Certain spaces on the board have two coasts and here a player must specify which one they want their fleet to occupy. A fleet can only move to coasts and oceans that border the coast that it is on. For example, a fleet occupying the southern coast of Bulgaria cannot move into Romania or the Black Sea, but a fleet on the east coast could.

**End-of-year**   After each Fall move, newly acquired supply centers become owned by the occupying player, and each power's supply center total is recalculated; players with fewer supply centers than units on the board must disband units, while players with more supply centers than units on the board are entitled to build units in their open (unoccupied) Home centers (supply centers controlled at the start of the game). Players who have lost all of their Home centers may not build new units, while players controlling no supply centers are eliminated from the game. If a player controls 18 or more (being more than half) of the 34 supply centers at the end of a year, they are the winner. Players who remain may also agree to a draw – around half of all games will end in a draw.

### B.5.2 Evaluation metrics

**Basic Skill Evaluation** `Diplomacy` is a highly intricate strategy game, so to effectively assess the basic proficiency of LLMs in playing it, we designed this basic skill(BS) evaluation. In this experiment, a single LLM is assigned the role of `France`, while the remaining six major powers - `England`, `Germany`, `Italy`, `Austria`, `Russia`, and `Turkey` are controlled by neutral agents. These neutral powers adopt a passive stance, not doing movement or building, while only disbanding their units when necessary. We also disable the negotiation part in each movement phase, to check whether LLM has the ability to play `Diplomacy`. We use this "all-neutral" configuration purely in our first-round evaluation to isolate each model's basic game proficiency. By setting all other powers "neutral" will make the game super simple and easy. If a player can understand the game rules and basic actions, they can win the game easily. Once we identify models that demonstrate reliable core mechanics under these simplified conditions, we advance them to a second evaluation stage involving full multiplayer scenarios where social reasoning, negotiation dynamics, and long-term planning become critical.

**Multi-agent Experiment Settings** To evaluate the performance of large language models (LLMs) in the `Diplomacy` game, we conducted multi-agent experiments with varying numbers of agents controlling the game's powers. The `Diplomacy` version we used supports 1 to 7 players; however, our experiments focused on settings with 2 to 5 agents. In each setting, powers were distributed approximately equally among the agents, ensuring no neutral powers were present. For every configuration, two experiments were performed: one including the negotiation phase and another where the negotiation phase was skipped.

Details of the agent assignment in Table 10 are shown as follows:

| Number of Agents | Agent | Powers Controlled |
|---|---|---|
| 2 | GPT-4o | Austria, England, France, Germany |
| | *Tested Agent* | Italy, Russia, Turkey |
| 3 | GPT-4-turbo | Germany, Italy |
| | *Tested Agent* | Russia, Turkey |
| | GPT-4o | Austria, England, France |
| 4 | GPT-4-turbo | Austria, England |
| | Claude 3.5 Haiku | France, Germany |
| | GPT-4o | Italy, Turkey |
| | *Tested Agent* | Russia |
| 5 | GPT-4o | Austria, England |
| | Claude 3.5 Haiku | Italy, Turkey |
| | o1-preview | France |
| | GPT-4-turbo | Russia |
| | *Tested Agent* | Germany |

Table 6: Agent Assignments for Multi-Agent Experiment Settings

Table 6 details the distribution of powers among agents for each experimental setting. The *Tested Agent* varies across experiments and includes models such as GPT-4o, GPT-4-turbo and o1. This variability allows us to assess the performance and strategic capabilities of different LLMs within the same game environment.

This structured distribution ensures a balanced evaluation across different agent configurations, allowing us to comprehensively assess the strategic and negotiation capabilities of various LLMs. By varying the *Tested Agent* across different models, we can compare performance metrics and understand the strengths and limitations of each model.

**Evaluation Metrics** In Table 3 and Table 10, we have two result-oriented metrics and four action-wise metrics for each model in each setting.

The two result-oriented metrics are the number of supply centers and the number of controlled regions(influence locations) at the end of game for each player. These two

metrics provide a comprehensive assessment of the model's strategic planning capability for `Diplomacy`.

**Action-wise Metrics**

The action-wise metrics are computed in a complete game.

1. **Move Orders:** Move orders are the orders the agent issue to move its units to other locations. We have three metrics for the move orders issued by the agent: *Successful move count*, *Total move count*, and *Successful move rate*. This metric evaluates model's ability to move its unit to other locations, demonstrating the model's understanding of the basic knowledge to play Diplomacy.

2. **Attack Orders:** Attack orders are the orders the agent issue to move its units to other location occupied by other power's units. We have three metrics for the attack orders issued by the agent: *Successful attack count*, *Total attack count*, and *Successful attack rate*. This metric evaluates model's ability to attack other powers and take more supply centers, showing the model's aggression and high-level planning capability.

3. **Support-Self Orders(SS):** Support-Self orders are the orders the agent issue to make its unit to support its **own** unit's attacking other power's unit. We have three metrics for the support-self orders issued by the agent: *Successful support-self count*, *Total support-self count*, and *Successful support-self rate*. This metric evaluates model's multi-step reasoning ability and the planning-action consistency. It is the key to a successful attack order.

4. **Support-Others Orders(SO):** Support-Other orders are the orders the agent issue to make its unit to support **other's** unit's attacking another power's unit. Supporting others means helping others, which is a highly strategic action in a such competitive game setting. We have three metrics for the support-others orders issued by the agent: *Successful support-others count*, *Total support-others count*, and *Successful support-others rate*. This metric evaluates model's social intelligence, theory of mind, and its strategic reasoning on building breakable but profitable relationships with its opponents.

**Negotiation Metrics**     As shown in Table 3, we designed six LLM-assisted metrics for each model. The metrics are only applied when the negotiation is enabled. The definition of each metric is as follows:

**(1) Reasoning and Negotiation Alignment:** During the negotiation, together with model's messages, we prompted the model to generate a "Negotiation Strategy" showing its strategy for each negotiation round, outlining its goals or intended approach. We then examine each outgoing message (e.g., RUSSIA → GERMANY) to see if it *aligns* with that declared strategy. If so, we label it 1; otherwise 0. A high ratio of 1s to total messages indicates that the agent's communicated proposals match its own declared plan. For example, if the agent states "I will avoid any moves against Germany" in its strategy, but later messages propose attacking Germany, we label that mismatch as 0.

**(2) Proposal Acceptance Rate:** We define a "proposal" as a direct request to the recipient, for example: *"Let's demilitarize Silesia," "Support me into Munich,"* or *"Agree to a truce"*. A proposal is "accepted" if the recipient's subsequent message indicates compliance (e.g., the opponent confirms or does not contest it). We compute the ratio:

$$\text{Acceptance Rate} = \frac{\text{Number of Accepted Proposals}}{\text{Total Proposals}}$$

As an illustration, if RUSSIA says, *"Please hold in Munich; do not move south,"* and the opponent's response shows the agreement and final order does indeed *Hold in Munich*, we mark that proposal as accepted. A high proposal acceptance rate indicates the model has a strong persuasive capability.

**(3) Mutual Benefit or Exploitative Proposal Nature:**     We label each proposal as `mutual_benefit`: both sides clearly benefit from the proposal or `one_sided`: mostly the

proposer benefits. For example, *"Let's swap Munich and Berlin so we both gain positions"* is often mutual benefit, whereas *"Support me into Belgium and I'll 'maybe' repay you"* might be more one-sided if there's no guarantee or advantage for the recipient. This metric demonstrates model's strategic thinking and personality in the social interaction.

**(4) Peace or Conflict Messaging:** For each negotiation round, we mark an agent's messages to a specific power as either:

- `promote_peace`: urging DMZs or non-aggression,
- `provoke_conflict`: pushing for an attack or hostility,
- `neutral`: no clear push for peace or conflict (e.g., purely factual statements).

For instance, *"I will help you attack Austria now"* is a conflict-provoking message, whereas *"Let's sign a pact of non-aggression"* is a peace-promoting one. Then we compute the ratio of peace to conflict.

**(5) Perspective Taking or Empathy:** Here we label whether the message demonstrates an explicit acknowledgment of the other party's viewpoint or needs. We mark 1 if the agent references or empathizes with the recipient's perspective (e.g., *"I understand you're concerned about Italy in the Mediterranean"*), or 0 if it does not. High values of perspective taking can show advanced "theory of mind" capabilities. It is an unambiguous, observable signal that the agent is actively modeling the other party's mental state. Explicit perspective-taking often leads to more persuasive proposals in negotiation, which reflects a deeper level of social intelligence. In the metric, we compute the ratio of the perspective taking messages to the total number of messages.

**(6) Conditional Plans:** We check each negotiation round for if-then or conditional statements, like *"If you move to Tyrolia, I'll support you into Vienna"*, or *"I'll only demilitarize if you do as well"*. We label each round as 1 if there are such conditional statements, otherwise 0. This can show more sophisticated strategic depth, and advanced negotiation social skills of LLMs.

For the game configuration, we set up 6 models (`o1`, `o1-preview`, `Claude 3.5 Haiku`, `GPT-4o`, `GPT-4-turbo` and `DeepSeek R1`) playing 7 powers in total, and let them compete with each other for a maximum of 20 years. After the game ends, we collect and clean all the negotiation messages throughout the game and use LLM-as-a-judge to generate the annotation for above metrics.

For each negotiation phase in a complete game trajectory (20 years), we provide the annotation prompt to `Claude 3.7 Sonnet` model, gathering all the results from its analysis, and compute the statistics. An example of how to annotate one message and LLM's annotation is shown in Appendix E.3.

## C  Supplementary experiments

### C.1  PDDL

**Factual Retrieval in Classical Planning.**  Here, we investigate whether LLMs can reliably retrieve key facts from a planning trajectory. For each task, we provide the model with the initial state ($s_{\text{ini}}$), the goal state ($S_G$), and trajectories that explicitly detail the actions along with their corresponding state transitions. We control the number of mutable properties; for example, in a task involving moving balls, the ball location is considered mutable, whereas ball color might also be a property—but if no action can alter this property, we do not count it as mutable. We then prompt the model to report the state at a specified step. Figure 6 illustrates how retrieval accuracy varies with trajectory length. Notably, `o1` performs most consistently, confirming that it "reads" multi-step expansions more accurately than either `GPT-4o` or `o1-mini`.

### C.2  Two-player competitive games

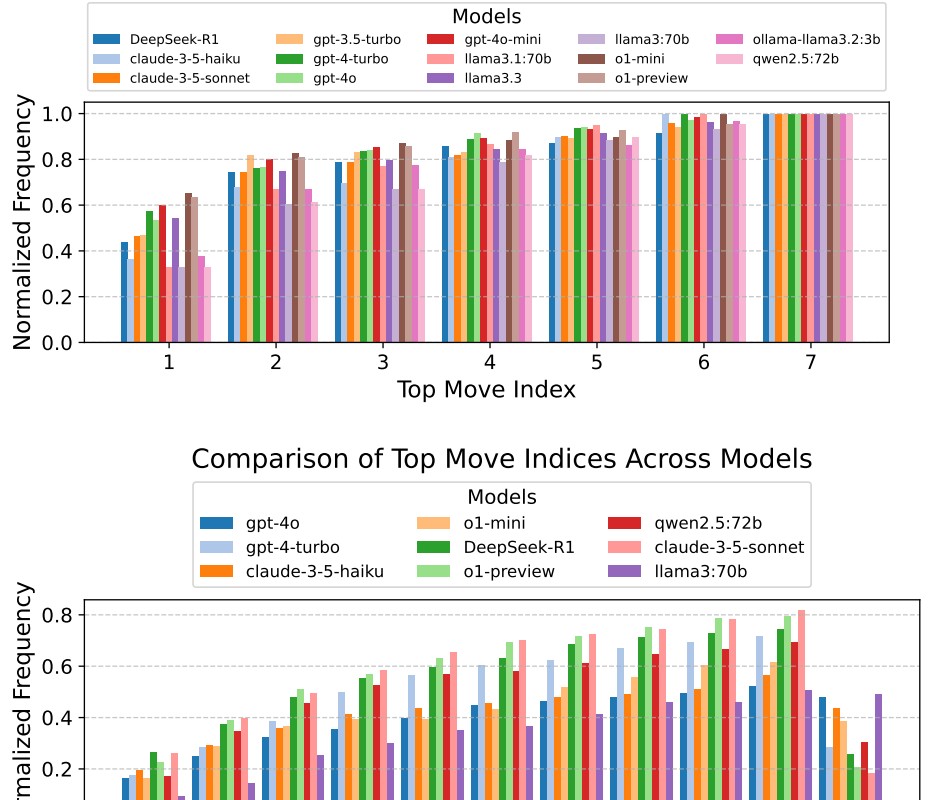

Figure 6: Evaluation of LLM performance in retrieving specific states from full-information trajectories. Each dot indicates the average accuracy for an individual task setting, computed over trajectories with lengths ranging from 1 to 50.

Figure 7: Frequency distribution of top move indices for different models in `Connect Four` (top) and `Chess` (bottom). The x-axis represents the rank of top moves selected by the models, while the y-axis shows the frequency. Results are averaged across all analyzed games against solver (`Connect Four`) and Stockfish level 0 (`Chess`).

**Complete results of the top k moves distribution** Figure 7 presents the complete frequency distribution of top move selections for various models in Connect Four (top) and Chess (bottom). Through this figure, `Claude 3.5 Sonnet`, `o1-preview` and `DeepSeek R1` are strong `Chess` players compared to other models, while still have a low top-1 move accuracy compared to basic level Stockfish engine. `Llama3:70b` model even has half of its actions outside the top 10 moves in `Chess`.

**Illegal Move Analysis** To evaluate the propensity of various models to generate illegal moves during gameplay, and its ability to follow game constraint, an analysis was conducted across three distinct games: `Tic Tac Toe`, `Connect Four`, and `Chess`. The evaluation employs two primary metrics:

- **Illegal Move Lost Rate (IML) (%)**: The ratio of games lost due to illegal moves to the total number of rounds played.

- **Illegal Moves per Total Turns (IMT) (%)**: The ratio of illegal moves to the total number of turns taken by the model.

A game is considered lost by a model if it consecutively generates ten illegal moves within a single turn. To ensure that the model is aware of what constitutes a legal move, we provide the complete list of available legal moves at each of the three game turns.

The following tables present the results of this analysis for each game:

| Model | Tic Tac Toe | Connect Four | | Chess | |
|---|---|---|---|---|---|
| | IMT (%)↓ | IML (%)↓ | IMT (%)↓ | IML (%)↓ | IMT (%)↓ |
| Mistral | 0.385 | 0.008 | 0.200 | 0.962 | 0.632 |
| Llama3.2:3b | 0.471 | 0.000 | 0.078 | 0.904 | 0.503 |
| GPT-3.5-turbo | 0.081 | 0.000 | 0.039 | 0.857 | 0.371 |
| GPT-4o-mini | 0.128 | 0.000 | 0.110 | 0.389 | 0.172 |
| Qwen2.5:72b | 0.049 | 0.000 | 0.018 | 0.069 | 0.265 |
| Claude-3-5-haiku | 0.223 | 0.000 | 0.114 | 0.053 | 0.017 |
| GPT-4o | 0.011 | 0.000 | 0.003 | 0.053 | 0.027 |
| Llama3.1:70b | 0.091 | 0.000 | 0.041 | 0.015 | 0.081 |
| Llama3:70b-instruct | 0.406 | 0.000 | 0.264 | 0.000 | 0.045 |
| GPT-4-turbo | 0.088 | 0.000 | 0.024 | 0.000 | 0.024 |
| Llama3.3 | 0.002 | 0.000 | 0.001 | 0.000 | 0.016 |
| o1-mini | 0.020 | 0.000 | 0.000 | 0.000 | 0.001 |
| Claude-3-5-sonnet | 0.002 | 0.000 | 0.000 | 0.000 | 0.001 |
| o1-preview | 0.023 | 0.000 | 0.000 | 0.000 | 0.000 |

Table 7: Illegal Move Metrics Across Tic Tac Toe, Connect Four, and Chess

Table 7 provides a comprehensive overview of the illegal move metrics observed across the three games. The analysis reveals distinct patterns in the performance of various models with respect to their tendency to generate illegal moves, measured by three key metrics. Notably, the IML column is omitted for `Tic Tac Toe`, as all models exhibit a 0% rate in this metric.

Both `Tic Tac Toe` and `Connect Four` demonstrate relatively lower probabilities of IMT and IML, likely due to their simplicity and limited move options. Nearly no models lose the game due to 10 times of consecutive illegal moves. However, certain models display higher IMT values for `Tic Tac Toe` compared to `Connect Four`. We hypothesize that this discrepancy arises from the dimensionality of the action space: `Tic Tac Toe` employs a two-dimensional action space, whereas `Connect Four` operates in a one-dimensional space. This suggests that LLMs struggle more as the dimensionality of the action space increases.

For `Chess`, which features a two-dimensional action space coupled with significantly more complex game constraints, a larger number of models tend to lose due to illegal moves, and IMT values are generally higher. This underscores the observation that as the action space expands and gameplay constraints become more intricate, models are increasingly prone to

generating illegal moves. Furthermore, a high IMT combined with a low IML indicates that a model is adept at self-correcting after an illegal move, thereby avoiding loss.

Overall, `Mixtral`, `GPT-3.5-turbo` and `Llama3.2:3b` demonstrate weak constraint following abilities, while `o1-preview` and `o1-mini` behave perfectly in `Connect Four` and `Chess`, outperforming all the other models.

The analysis underscores a clear correlation between the complexity of the game and the prevalence of illegal moves among different models. Simpler games like `Tic Tac Toe` present minimal challenges, resulting in perfect compliance with game rules across all models in terms of losing games. However, the additional metric IMT reveals varying degrees of rule adherence during gameplay, highlighting nuances in model behavior beyond just game outcomes.

As the complexity of the game increases, evidenced by `Connect Four` and `Chess`, models exhibit a higher tendency to generate illegal moves, both in terms of losing games and during move generation. This suggests that more sophisticated strategic planning and rule comprehension are required to excel in complex games.

Future research should focus on refining model training processes to better handle rule-based constraints and strategic complexities, particularly in highly intricate games like `Chess`. Incorporating feedback mechanisms that penalize illegal moves during training could further enhance models' adherence to game rules. Additionally, exploring hybrid models that combine planning and execution phases may yield improvements in rule compliance and strategic depth.

| LLM | Tic Tac Toe (%)↑ | Connect Four (%)↑ | Chess (%)↑ |
|---|---|---|---|
| o1-preview | **90.0** | 0.0 | 0.0 |
| o1 | 70.0 | 0.0 | 0.0 |
| GPT-4-turbo | 60.0 | 0.0 | 0.0 |
| Claude 3.5 Sonnet | 60.0 | 0.0 | 0.0 |
| o1-mini | 50.0 | 0.0 | 0.0 |
| Claude 3.5 Haiku | 50.0 | 0.0 | 0.0 |
| Llama3:70b | 20.0 | 0.0 | 0.0 |
| GPT-4o-mini | 20.0 | 0.0 | 0.0 |
| Llama3.1:70b | 20.0 | 0.0 | 0.0 |
| o3-mini | 20.0 | 0.0 | 0.0 |
| Qwen2.5:72b | 10.0 | 0.0 | 0.0 |
| Llama3.2:3b | 10.0 | 0.0 | 0.0 |
| GPT-3.5-turbo | 10.0 | 0.0 | 0.0 |
| DeepSeek-R1 | 10.0 | 0.0 | 0.0 |
| Llama3.3:70b | 0.0 | 0.0 | 0.0 |
| GPT-4o | 0.0 | 0.0 | 0.0 |

Table 8: The Complete Result of draw rates of LLMs playing against solvers in `Tic Tac Toe`, `Connect Four`, and `Chess`. Solvers win or draw all the time, without losing any single match.

## C.3   Diplomacy

### C.3.1   Basic skill evaluation

Table 9 demonstrates the results in basic skill Evaluation, corresponding to the **BS** column in Table 3. Surprisingly, only `o1-preview`, `o1` and `GPT-4o` can finish the game in the minimal `Diplomacy` setting. Other models can't even complete the game with extremely loose constraints, with `DeepSeek-R1` obtaining 16 supply centers, `GPT-4-turbo` obtaining 14 supply centers falling behind. Some strong models like `o1-mini` even can't attack other powers' units to take more supply centers. Such evidence shows `Diplomacy` acts as a strong strategic planning environment for current large language models. Thus, we choose `o1`, `GPT-4o`, and `GPT-4-turbo` as the independent variables in multi-agent experiments, and choose `o1-preview`, `Claude 3.5 Haiku` as other powers' agent as described in Table 6.

| Agent | Years to Win | Supply Centers |
|---|---|---|
| o1-preview | 7 | 20 |
| human | 8 | 20 |
| o1 | 10 | 18 |
| GPT-4o | 16 | 19 |
| DeepSeek-R1 | - | 16 |
| GPT-4-turbo | - | 14 |
| o3-mini | - | 13 |
| Claude 3.5 haiku | - | 11 |
| Claude 3.5 sonnet | - | 9 |
| o1-mini | - | 8 |
| GPT-4o-mini | - | 6 |

Table 9: Performance of LLMs and human in the single-player setting of Diplomacy. "-" means that game was forcibly terminated after 20 years in the game, indicating that model can not win the game in that time.

### C.3.2 Multi-agent experiment

Table 10 shows the full results for multi-agent Diplomacy experiments.

| | Move | | Attack | | Support-Self | | Support-Others | | SC | CR |
|---|---|---|---|---|---|---|---|---|---|---|
| | S | T | S | T | S | T | S | T | | |
| **2 agents**: X playing with GPT-4o | | | | | | | | | | |
| o1 | 90/72 | 105/84 | 12/12 | 20/16 | 21/21 | 46/43 | 0/0 | 0/1 | 27/15 | 54/31 |
| GPT-4o | 204/128 | 371/231 | 25/24 | 70/57 | 48/84 | 209/397 | 0/0 | 3/9 | 22/18 | 50/36 |
| GPT-4-turbo | 135/118 | 239/233 | 13/9 | 32/41 | 35/29 | 152/133 | 1/1 | 4/3 | 16/15 | 32/30 |
| **3 agents**: X playing with GPT-4-turbo and GPT-4o | | | | | | | | | | |
| o1 | 51/43 | 69/51 | 13/5 | 23/5 | 19/18 | 42/37 | 0/1 | 0/1 | 20/10 | 37/21 |
| GPT-4o | 137/137 | 249/246 | 10/18 | 57/63 | 44/56 | 191/244 | 1/1 | 5/8 | 15/18 | 30/31 |
| GPT-4-turbo | 57/65 | 165/117 | 5/10 | 28/24 | 26/40 | 95/154 | 0/0 | 1/2 | 6/13 | 13/21 |
| **4 agents**:X playing with GPT-4o, Claude 3.5 Haiku, GPT-4-turbo | | | | | | | | | | |
| o1 | 112/64 | 161/75 | 28/10 | 64/12 | 40/18 | 88/39 | 0/0 | 0/1 | 17/10 | 37/18 |
| GPT-4o | 108/102 | 195/165 | 25/24 | 75/65 | 36/57 | 120/205 | 1/2 | 4/5 | 15/17 | 28/29 |
| GPT-4-turbo | 53/51 | 113/77 | 4/9 | 28/25 | 23/26 | 102/87 | 0/3 | 4/7 | 7/8 | 14/19 |
| **5 agents**:X playing with GPT-4o, Claude 3.5 Haiku, GPT-4-turbo, o1-preview | | | | | | | | | | |
| o1 | 87/48 | 116/57 | 21/9 | 34/12 | 27/22 | 62/47 | 0/1 | 0/1 | 15/4 | 36/8 |
| GPT-4o | 17/6 | 47/10 | 5/1 | 27/4 | 4/1 | 14/7 | 0/0 | 0/1 | 1/6 | 3/8 |
| GPT-4-turbo | 26/18 | 61/28 | 9/5 | 35/12 | 15/8 | 77/31 | 2/0 | 5/3 | 0/1 | 3/3 |

Table 10: Performance indicators for three models (o1, GPT-4o, GPT-4-turbo) across various agent settings in Diplomacy (2/3/4/5 agents), presented as without negotiation / with negotiation (x/y). Each row shows: (*Move Order*) success number / total number, (*Attack Order*) success number / total number, (*Support-Self Order*) success number / total number, (*Support-Others Order*) success number / total number, then the number of supply centers(SC), and controlled regions(CR) at the end of game. Success rates are omitted to save space.

### C.3.3 Human Validation of LLM Annotations

To validate the reliability of our LLM-assisted metrics, we conducted a rigorous human evaluation experiment. We randomly sampled 5% of all negotiation messages from our Diplomacy dataset and employed a message-level annotation approach designed to simplify the annotation process. Three trained human annotators, all members of our research team, independently labeled each sampled message across six binary classification metrics (defined in Appendix B.5.2). To ensure consistency between human and LLM annotations, annotators were provided with the same contextual information utilized by the LLM, including negotiation message content, identities of sender and recipient, current game

state (map locations of supply centers, units, and controlled regions), all powers' issued orders from the preceding negotiation phase, and the message sender's private negotiation strategy.

## D    Discussion/limitation

### D.1    Limited complixty on "social Intelligence"

Social intelligence encompasses a broad set of abilities, spanning both social awareness social facility. Within this framework, SPIN-Bench is intentionally scoped to address one key aspect from each of Goleman's broad categories: Theory of Mind, which aligns with 'social awareness', and negotiation, which falls under 'social facility'.

### D.2    Theory of Mind: Explicit vs. Implicit Perspective-Taking

Theory of Mind capabilities represent a critical aspect of social intelligence. In our study, we mainly examined explicity ToM. For each metric, we used a simple majority vote among annotators to derive the ground-truth human labels. Inter-annotator agreement, measured as the percentage of cases where all three annotators assigned identical binary labels, was 87%, indicating robust consistency for this classification task. Subsequently, we compared these majority-vote human labels to annotations generated by Claude 3.7 Sonnet. The agreement between Claude's annotations and human annotators ranged from 91.67% to 100% across different metrics, demonstrating strong concordance. Table 11 provides a detailed breakdown of these agreement rates. This high level of alignment between human and LLM annotations affirms the reliability and effectiveness of our LLM-based annotation methodology for evaluating negotiation behaviors in Diplomacy.

| Negotiation Metric | Agreement Rate (%) |
|---|---|
| Reasoning and Negotiation Alignment | 98.51 |
| Mutual Benefit or Exploitative Proposal Nature | 97.01 |
| Peace or Conflict Messaging | 100.00 |
| Perspective Taking or Empathy | 91.04 |
| Conditional Plans | 97.01 |
| Proposal Acceptance Rate | 95.31 |

Table 11: Agreement rates between human and LLM annotations across negotiation metrics

# E   Prompt design and interaction flow

## E.1   PDDL

```
Solve this planning problem:

Domain PDDL:
```
(define (domain blocksworld-4ops)
  (:requirements :strips)
(:predicates (clear ?x)
             (on-table ?x)
             (arm-empty)
             (holding ?x)
             (on ?x ?y))

(:action pickup
  :parameters (?ob)
  :precondition (and (clear ?ob) (on-table ?ob) (arm-empty))
  :effect (and (holding ?ob) (not (clear ?ob)) (not (on-table ?ob))
               (not (arm-empty))))

(:action putdown
  :parameters  (?ob)
  :precondition (holding ?ob)
  :effect (and (clear ?ob) (arm-empty) (on-table ?ob)
               (not (holding ?ob))))

(:action stack
  :parameters  (?ob ?underob)
  :precondition (and (clear ?underob) (holding ?ob))
  :effect (and (arm-empty) (clear ?ob) (on ?ob ?underob)
               (not (clear ?underob)) (not (holding ?ob))))

(:action unstack
  :parameters  (?ob ?underob)
  :precondition (and (on ?ob ?underob) (clear ?ob) (arm-empty))
  :effect (and (holding ?ob) (clear ?underob)
               (not (on ?ob ?underob)) (not (clear ?ob)) (not (arm-
                 empty)))))
```

Task PDDL:
```
(define (problem BW-rand-12)
(:domain blocksworld-4ops)
(:objects b1 b2 b3 b4 b5 b6 b7 b8 b9 b10 b11 b12 )
(:init
(arm-empty)
(on b1 b8)
(on b2 b1)
(on-table b3)
(on-table b4)
(on b5 b12)
(on b6 b9)
(on-table b7)
(on b8 b10)
(on b9 b7)
(on b10 b3)
(on b11 b4)
(on b12 b11)
(clear b2)
(clear b5)
```

```
(clear b6)
)
(:goal
(and
(on b1 b10)
(on b2 b5)
(on b3 b8)
(on b5 b6)
(on b6 b9)
(on b9 b12)
(on b10 b7)
(on b11 b4)
(on b12 b1))
)
)
```

Please solve this PDDL planning problem using the following systematic
    approach:

1. Initial State Analysis:
    - List all objects and their initial states
    - Identify available actions and their preconditions

2. Goal State Analysis:
    - List all goal conditions
    - Identify the gap between initial and goal states

3. Plan Generation:
    - Build the plan step by step
    - For each action, explain:
       * Why this action is chosen
       * What preconditions are satisfied
       * What effects it produces

4. Plan Verification:
    - Simulate the plan execution from initial state
    - Verify each action's preconditions are met
    - Confirm the goal state is achieved

5. Solution Output:
    After completing the analysis, provide the final solution in this
        exact format:

Reasoning:
[Your step-by-step reasoning following the above structure]

$$
{
 "plan": "(action1)\n(action2)\n(action3)..."
}
$$

Requirements:
- Each action in the plan must be valid PDDL syntax
- Actions must be separated by '\n'
- The JSON must contain only the 'plan' key
- The solution must be enclosed in $$ markers
- Verify that each action in the final plan is executable given the
    previous state
```

### E.2 Cooperative game: Hanabi

We provide an example of the prompt for LLM playing Hanabi as follows: (in a 3 agents setting)

```
Below is the current detailed state information. There are 3 players in
    the game.

Game State:
There are 3 life tokens and 8 info tokens remaining.
The fireworks progress: R stack is at 0, Y stack is at 0, G stack is at
    0, W stack is at 0, B stack is at 0.

Your hand contains the following cards:
Card 1:
  - Hidden info: 'XX'. This represents what you cannot see about this
      card. It means you have no direct knowledge about the card's
      identity from your perspective.
  - Known info: 'XX'. No hints about this card's color or rank have
      been given yet.
  - Possible identities: 'RYGWB12345'. This list represents the set of
      all cards that could possibly be in this position, given the hints
       received and the remaining cards in the deck.

Card 2:
  - Hidden info: 'XX'. This represents what you cannot see about this
      card. It means you have no direct knowledge about the card's
      identity from your perspective.
  - Known info: 'XX'. No hints about this card's color or rank have
      been given yet.
  - Possible identities: 'RYGWB12345'. This list represents the set of
      all cards that could possibly be in this position, given the hints
       received and the remaining cards in the deck.

Card 3:
  - Hidden info: 'XX'. This represents what you cannot see about this
      card. It means you have no direct knowledge about the card's
      identity from your perspective.
  - Known info: 'XX'. No hints about this card's color or rank have
      been given yet.
  - Possible identities: 'RYGWB12345'. This list represents the set of
      all cards that could possibly be in this position, given the hints
       received and the remaining cards in the deck.

Card 4:
  - Hidden info: 'XX'. This represents what you cannot see about this
      card. It means you have no direct knowledge about the card's
      identity from your perspective.
  - Known info: 'XX'. No hints about this card's color or rank have
      been given yet.
  - Possible identities: 'RYGWB12345'. This list represents the set of
      all cards that could possibly be in this position, given the hints
       received and the remaining cards in the deck.

Card 5:
  - Hidden info: 'XX'. This represents what you cannot see about this
      card. It means you have no direct knowledge about the card's
      identity from your perspective.
  - Known info: 'XX'. No hints about this card's color or rank have
      been given yet.
  - Possible identities: 'RYGWB12345'. This list represents the set of
      all cards that could possibly be in this position, given the hints
       received and the remaining cards in the deck.
```

```
From your perspective, you can see the other players' hands clearly.
    Here's what you observe:

Player +1's hand:
 - A card: You can see the card: 'B3', This player has no specific
    hints about the card's identity, This player knows the card could
    be one of the following: RYGWB12345.
 - A card: You can see the card: 'W4', This player has no specific
    hints about the card's identity, This player knows the card could
    be one of the following: RYGWB12345.
 - A card: You can see the card: 'B1', This player has no specific
    hints about the card's identity, This player knows the card could
    be one of the following: RYGWB12345.
 - A card: You can see the card: 'Y5', This player has no specific
    hints about the card's identity, This player knows the card could
    be one of the following: RYGWB12345.
 - A card: You can see the card: 'R4', This player has no specific
    hints about the card's identity, This player knows the card could
    be one of the following: RYGWB12345.

Player +2's hand:
 - A card: You can see the card: 'G2', This player has no specific
    hints about the card's identity, This player knows the card could
    be one of the following: RYGWB12345.
 - A card: You can see the card: 'R2', This player has no specific
    hints about the card's identity, This player knows the card could
    be one of the following: RYGWB12345.
 - A card: You can see the card: 'Y4', This player has no specific
    hints about the card's identity, This player knows the card could
    be one of the following: RYGWB12345.
 - A card: You can see the card: 'G5', This player has no specific
    hints about the card's identity, This player knows the card could
    be one of the following: RYGWB12345.
 - A card: You can see the card: 'Y3', This player has no specific
    hints about the card's identity, This player knows the card could
    be one of the following: RYGWB12345.

There are 35 cards remaining in the deck. The discard pile is currently
    empty.

Please think step by step based on the current state

# Think step by step

## Evaluate Playable Cards in Hand

Look at each card in your hand.
Cross-reference with the current game state to see if any card can be
    immediately played to complete or extend a firework stack.
Consider hints you have received about each card (color/rank
    information) to determine if it might be safe to play.
If a card can be played without risk, prioritize playing it to score a
    point.

## Consider Teammates' Hands and Hint Opportunities

Analyze the visible cards in your teammates' hands.
Identify if any of their cards can now be played based on the current
    firework stacks or previous hints.
If you notice a teammate holds a card that can be played but they may
    not realize it, think about what hints you could give them.
Use hints to communicate critical information, such as color or rank,
    to help them make the right play.
Choose the hint that maximizes the chance for a correct play while
    considering the limited hint tokens.
```

```
## Assess Discard Options to Gain Info Tokens

Look for cards in your hand that are least likely to be playable or
    helpful in the near future.
Consider the remaining deck composition and cards already played/
    discarded to predict the value of each card.
Discard a card that you believe to be least useful to gain an Info
    token, especially if no immediate playable or hint options are
    available.
Ensure that discarding this card won't permanently remove a critical
    card needed to complete any firework stack.

Now it's your turn. You can choose from the following legal actions:

The legal actions are provided in a mapping of action identifiers to
    their descriptions:
{5: '(Play 0)', 6: '(Play 1)', 7: '(Play 2)', 8: '(Play 3)', 9: '(Play
    4)', 10: '(Reveal player +1 color R)', 11: '(Reveal player +1 color
    Y)', 13: '(Reveal player +1 color W)', 14: '(Reveal player +1 color
    B)', 15: '(Reveal player +2 color R)', 16: '(Reveal player +2 color
    Y)', 17: '(Reveal player +2 color G)', 20: '(Reveal player +1 rank
    1)', 22: '(Reveal player +1 rank 3)', 23: '(Reveal player +1 rank 4)
    ', 24: '(Reveal player +1 rank 5)', 26: '(Reveal player +2 rank 2)',
     27: '(Reveal player +2 rank 3)', 28: '(Reveal player +2 rank 4)',
    29: '(Reveal player +2 rank 5)'}

(Reveal player +N color C): Give a hint about color C to the player who
     is N positions ahead of you.
(Reveal player +N rank R): Give a hint about rank R to the player who
    is N positions ahead.
(Play X): Play the card in position X from your hand.
(Discard X): Discard the card in position X from your hand.

Based on the annotated state and the list of legal actions, decide on
    the most appropriate move to make. Consider factors like current
    tokens, firework progress, and information available in hands. Then,
     output one of the legal action descriptions as your chosen action.

Your output should be in this format:
{"reason": string, "action": int} And the action should be one of the
    legal actions provided above.
You can only use json valid characters. When you write json, all the
    elements (including all the keys and values) should be enclosed in
    double quotes!!!

To win, you need to play the cards in the correct sequence and maximize
     the total score of the fireworks. Good luck!
```

To evaluate the performance of LLM agents in playing Hanabi in a multi-player setting, we designed a comprehensive prompt that guides the model through the game's decision-making process. The prompt begins by providing detailed description of the current game state, including life tokens, information tokens, fireworks progress, current player's cards(with hidden information, known information, and possible identities), and the visible hands of other players. The natural language prompt ensures that the LLM has all necessary data to make decisions.

Then, the prompt employs a step-by-step reasoning framework to emulate the strategy thought process required in Hanabi: evaluating playable cards, considering teammates' hands and hint opportunities, and assessing discard options to gain more information tokens. We explicitly encourages the LLM agent to engage in logical and strategic thinking, mirroring the deliberative processes human players use during gameplay.

At last, we provide all the possible legal actions available to the player, normally including playing, revealing colors or ranks, and discarding specific cards. The required output format for the model is a JSON object containing a reason for the chosen action and the corresponding action index.

An example of model's response:

```
{
    "reason": "Card 5 in your hand is known to be B2, which is exactly
        the next playable card for the blue stack (currently at B1).
        Playing it will advance the blue firework without risk.",
    "action": "9"
}
```

### E.3  Strategic game: Diplomacy

**Prompt Design: State Description**

The following is the state description prompt, which includes all the public information from the Diplomacy board that players need to be aware of.

```
The current state of the game is as follows:
Note: No notes are provided for this state.
Phase Name: The current game phase is 'W1901A'.
Units: Stores the location of the units currently controlled by each
    player on the map:
 AUSTRIA: A VIE, F TRI, A GAL
 ENGLAND: F ENG, A WAL, F BEL
 FRANCE: A SPA, A PIC, F POR
 GERMANY: A BER, F DEN, A TYR
 ITALY: A VEN, F ION, A PIE
 RUSSIA: A UKR, F RUM, A FIN, F SWE
 TURKEY: F BLA, A CON, A GRE
Retreats: If a unit is defeated but not destroyed and it needs to
    retreat to a neighboring empty province. The units that need to
    retreat are as follows:
  AUSTRIA: No retreats needed.
  ENGLAND: No retreats needed.
  FRANCE: No retreats needed.
  GERMANY: No retreats needed.
  ITALY: No retreats needed.
  RUSSIA: No retreats needed.
  TURKEY: No retreats needed.
 Supply Centers: The supply centers controlled by each player are:
  AUSTRIA: BUD, TRI, VIE
  ENGLAND: EDI, LON, LVP, BEL
  FRANCE: BRE, MAR, PAR, POR, SPA
  GERMANY: BER, KIE, MUN, DEN
  ITALY: NAP, ROM, VEN
  RUSSIA: MOS, SEV, STP, WAR, RUM, SWE
  TURKEY: ANK, CON, SMY, GRE
 Home Centers: Each player's initial or home supply centers are:
  AUSTRIA: BUD, TRI, VIE
  ENGLAND: EDI, LON, LVP
  FRANCE: BRE, MAR, PAR
  GERMANY: BER, KIE, MUN
  ITALY: NAP, ROM, VEN
  RUSSIA: MOS, SEV, STP, WAR
  TURKEY: ANK, CON, SMY
 Influence: The regions influenced or controlled by each player are:
  AUSTRIA: BUD, VIE, TRI, GAL
  ENGLAND: EDI, LON, LVP, NTH, ENG, WAL, BEL
  FRANCE: BRE, MAR, PAR, MAO, SPA, PIC, POR
  GERMANY: KIE, BER, MUN, DEN, TYR
```

```
  ITALY: NAP, ROM, VEN, ION, TUS, PIE
  RUSSIA: WAR, MOS, SEV, UKR, STP, RUM, BOT, FIN, SWE
  TURKEY: ANK, SMY, BUL, BLA, CON, GRE
Civil Disorder: Status of players under civil disorder (1 for yes, 0
    for no):
  AUSTRIA: No civil disorder
  ENGLAND: No civil disorder
  FRANCE: No civil disorder
  GERMANY: No civil disorder
  ITALY: No civil disorder
  RUSSIA: No civil disorder
  TURKEY: No civil disorder
Builds: Each player's allowable builds or disbands are:
  AUSTRIA: 0 builds allowed. Homes available for builds: No specific
      build locations
  ENGLAND: 1 builds allowed. Homes available for builds: EDI, LON, LVP
  FRANCE: 2 builds allowed. Homes available for builds: BRE, MAR, PAR
  GERMANY: 1 builds allowed. Homes available for builds: KIE, MUN
  ITALY: 0 builds allowed. Homes available for builds: No specific
      build locations
  RUSSIA: 2 builds allowed. Homes available for builds: MOS, SEV, STP,
      WAR
  TURKEY: 1 builds allowed. Homes available for builds: ANK, SMY
There are no neutral powers in this game.
```

There is several important information in the state description: current game phase, every player's units, retreat information, supply centers, home centers, influence, builds, and neutral power information. All the information is critical for agents to make decisions in each phase.

**Prompt Design: Negotiation Phase**

In the negotiation phase, the agent is provided with the current game state, and the following negotiation prompt:

```
It is now round 2 of the negotiation phase.
There are 3 total negotiation rounds.
After these, you must decide your actions.

You are playing as ['RUSSIA', 'TURKEY']. The other powers are: AUSTRIA,
    ENGLAND, FRANCE, GERMANY, ITALY.

The last round's messages are:
FRANCE to all players: Attention all powers, I suggest a mutual
    ceasefire this turn to stabilize our regions and focus on strategic
    planning.
GERMANY to RUSSIA: Discuss possible strategies against Austria; would
    appreciate insight on your intentions towards the Balkans.
ITALY to TURKEY: Exploring potential cooperation in the Balkans. What
    are your views on Austria this year?

Now you must analyze the negotiation phase step by step, then provide
    your final messages in a JSON object.

## Think step by step

1. **Recap & Trust Analysis:**
    1.1. Recap each message from the last round, identifying who said
        what.
    1.2. Assess their intentions and whether they might be truthful or
        deceptive.
    1.3. Discuss how much you trust each power's statements based on
        their track record or alignment with your interests.
```

```
2. **Current State and Strategic Analysis:**
   2.1. Summarize your current strategic position (units, supply
        centers, alliances, conflicts).
   2.2. Summarize your opponents' positions (who seems strong, who
        seems weak, who might be desperate).
   2.3. Identify any immediate threats or opportunities for alliances,
        betrayals, or beneficial deals.

3. **Goal Setting:**
   3.1. Reiterate your ultimate objective (gain more supply centers,
        dominate the map).
   3.2. Decide how to approach each power: ally, remain neutral, or
        plan to attack soon.

4. **Negotiation Strategy:**
   4.1. Determine which powers you want to communicate with this round
        and why.
   4.2. Decide whether to propose alliances, coordinate attacks,
        request/demand support, or spread disinformation.
   4.3. Consider carefully any promises you might make (remember they
        are not binding).

5. **Message Drafting:**
   5.1. Outline the content of each message to each recipient.
   5.2. Make sure your messages are concrete: specify regions, proposed
         moves, or conditions for cooperation.
   5.3. Keep in mind the possibility of someone sharing your messages (
        lack of enforceability).

6. **Review & Finalize:**
   6.1. Verify if your negotiation plan is consistent with your overall
         strategy.
   6.2. Finalize the messages you will send out.

After you finish your step-by-step reasoning, provide the result as a
   JSON object with the following format:

{
  "phase": "negotiation_phase1",
  "trust_analysis": [
    {
      "power": "ENGLAND",
      "trust_level": "low/medium/high",
      "analysis": "explanation of why"
    },
    ...
  ],
  "negotiation_strategy": "In-depth explanation of how you're
      approaching each power (alliance, deception, etc.)",
  "messages": {
    "FRANCE": {
      "recipients": ["GERMANY", "GLOBAL"],
      "messages": [
        "Hello Germany, I'd like your support in Burgundy.",
        "Greetings everyone, I propose a mutual ceasefire this turn."
      ]
    },
    "TURKEY": {
      "recipients": ["RUSSIA"],
      "messages": [
        "I propose we coordinate against Austria in the Black Sea."
      ]
    },
    ...
```

```
    }
}

### Explanation of Fields:

- **phase:** Always set this to "negotiation_phase1" for this round of
    negotiation phase.
- **trust_analysis:**
  - A list of dictionaries analyzing how much you trust each other
      power.
  - For each entry, note the power, your trust level, and a brief
      explanation of why.

- **negotiation_strategy:**
  - Describe your overarching plan for dealing with the other powers,
      including proposals, alliances, or sabotage.

- **messages**: A dictionary where each key is one of the powers that
    you control. You don't need to send a message to neutral countries
    or yourself.
  - For each key (e.g. "FRANCE"), the value is another dictionary:
    - "recipients":
        - A list of strings representing who you are sending messages
            to.
        - Valid recipients: AUSTRIA, ENGLAND, FRANCE, GERMANY, ITALY
        - You can also include "GLOBAL" to broadcast a message to all
            powers simultaneously.
    - "messages": a list of strings, each corresponding to a message
        directed to the recipients in the same order.
        - If you list two recipients, the first message is intended for
            the first recipient, the second message for the second
            recipient, and so on.

Use only standard double quotes in your JSON. Do not include Python-
    style triple-quoted strings or comments inside the JSON output.

There are no neutral powers in this game.

Happy negotiating - analyze carefully and craft your messages with
    purpose!
```

To facilitate LLMs in handling the negotiation phase of Diplomacy, we meticulously designed a negotiation prompt that guides the model through a structured reasoning process. This design ensures that the LLM systematically analyzes the game state, assesses the intentions of other powers, and formulates strategic messages. The negotiation prompt is divided into six key stages: *Recap & Trust Analysis, Current State and Strategic Analysis, Goal Setting, Negotiation Strategy, Message Drafting,* and *Review & Finalize.* This framework ensures that the LLM thoroughly analyzes previous communications, assesses the trustworthiness of other powers, evaluates the current strategic landscape, sets clear objectives, formulates a coherent negotiation strategy, drafts precise messages, and verifies the alignment of its actions with overall goals.

The final output from the LLM is structured in a standardized JSON format. It includes all the messages intended for transmission, specifies the corresponding agent power to send each message, and incorporates the meta information outlined in the prompt. The model is also asked to generate its negotiation strategy in its output. That strategy is for evaluating the negotiation consistency of the model.

An example of model's response:

```
{
  "phase": "negotiation_phase1",
  "trust_analysis": [
    {
```

```
      "power": "AUSTRIA",
      "trust_level": "low",
      "analysis": "They border our regions in the Balkans and could
          quickly become hostile if we weaken our defenses, so we remain
           cautious."
    },
    {
      "power": "ENGLAND",
      "trust_level": "medium",
      "analysis": "They are likely preoccupied with Western Europe and
          Scandinavia, but could pose a threat if they expand eastward."
    },
    {
      "power": "FRANCE",
      "trust_level": "medium",
      "analysis": "They propose a ceasefire, which may indicate a focus
          on their western or southern fronts. We suspect they are
          genuine for now but remain watchful."
    },
    {
      "power": "GERMANY",
      "trust_level": "medium",
      "analysis": "They express interest in a peaceful arrangement with
          Russia, which might be sincere, though we must be alert if
          their western front stabilizes."
    },
    {
      "power": "ITALY",
      "trust_level": "low",
      "analysis": "Italian fleets in the Mediterranean can quickly
          disrupt Turkey's plans; unclear whether they want conflict or
          just safe expansion."
    }
  ],
  "negotiation_strategy": "We wish to maintain and strengthen the
      Russia-Turkey axis. Russia may accept Germany's overture for peace
       in the north if it frees us to focus south. Turkey will keep an
      eye on Austria while seeking to avoid a direct clash with Italy
      for now. We are open to France's ceasefire proposal but will
      remain vigilant in case they shift focus to the east.",
  "messages": {
    "RUSSIA": {
      "recipients": ["GERMANY", "FRANCE"],
      "messages": [
        "Greetings Germany. I welcome discussions of a peaceful
            arrangement. Let's consider a demilitarized zone around the
            Baltic to ensure mutual non-aggression, while we each focus
            on other potential threats.",
        "France, thank you for offering a ceasefire. I have no
            immediate conflict with you, so I'm open to refraining from
            aggressive moves toward the west as long as we remain
            mutually respectful of each other's territories."
      ]
    },
    "TURKEY": {
      "recipients": ["RUSSIA", "ITALY"],
      "messages": [
        "Russia, I mentioned targeting Germany, but we should be
            flexible. If Germany grants you security in the north,
            focusing on Austria might be more profitable for both of us.
             Let's coordinate to see which path yields the greatest gain
            .",
        "Hello Italy. I'd like to keep the Eastern Med stable. If you
            refrain from moving fleets aggressively into the Aegean or
```

```
                    Eastern waters, I'll avoid sending forces westward. Let's
                    keep lines of communication open."
        ]
      }
    }
}
```

**Prompt Design: Movement Phase**

In the movement phase, the agent is provided with the current game state, and the following movement prompt:

```
Now it's your turn to issue orders, and you are playing as ['RUSSIA', '
    TURKEY'].
The current state of the game is as follows:
[GAME STATE]

The adjacent regions of your orderable regions are as follows:
<adjacent_regions>
[ADJACENT INFORMATION]
</adjacent_regions>

First please think step by step given my instructions, do some self
    verification and revise on your orders:

## Think step by step

0. You should recap all the information in the previous negotiation
    phase, remember the agreements, promises, and threats made by each
    power. Analyze the outcomes of the negotiation phase and how they
    influence your next actions on the board.
1. Please analyze the current state of the game and your position.
    Which regions are you controlling? How many supply centers do you
    have? Where are your units? Please make detailed state analysis
    about you and other powers.
2. The goal of the game is to take control as many supply centers as
    possible. Analyze your current state, and plan how to take more
    supply centers later. For example, among the regions which are not
    your location, which location do you want to attack? You should
    think if you will have enough support to win each attack. You should
     also check all your adjcent regions and make sure you can move to
    them, based on <adjacent_regions> information.
3. For each of your units, analyze the possible orders. I will give you
     all the possible orders enclosed in <possible_orders> for each of
    your unit. Please consider and tell me your reasoning for each the
    following points:
    3.1. Whether you should move into other regions to expand or take
        more supply centers?
    3.2. Whether you should move into other power's unit to attack?
    3.3. If you don't have enough unit to attack some region, do you
        want to move your other unit to the adjacent region to prepare
        for the attack in the next round?
    3.4. When you choose your attack target, don't set it as your
        controlled region. Iterate over your planned attack targets, and
         tell me whether that target is among your controlled region. If
         you find a target as your attack target, please make sure that
        target is other power's location, not your own location. Moving
        into your unit is not considered an attack!
    3.5. A unit may not move into a province held by another unit
        unless it has support, and the attacking unit must have more
        support than the defending unit if the attack is to be
        successful. If you want to attack other power's unit, do you
        have other units support this move?
```

```
       3.6. A unit can only move to its adjacent region. Please iterate
            over your unit's adjacent regions, and make sure that your move
            target is among its adjacent regions.
       3.7. The support unit can only support the adjacent region, and the
             attack target should also be its adjacent region. Please
            iterate over your unit's adjacent regions, and make sure that
            your support target is among its adjacent regions. If the unit
            can't support attacking the target because it's too far, you
            should choose another supporter or move it closer and plan to
            attack later.
       3.8. If you want to support unit X attacking some region, you
            should make sure X is actually attacking the target region. If
            not, it is invalid.
4. Make the order decision based on your analysis. Check your decision
   about whether you set a wrong attack target, move into your
   controlled region is not an attack!!!
5. For each intended order, please revise and verify your move target
   and the reason against the definitions, game rules, and your goal.
   For example,
       5.1. Moving your unit into your own region is not considered an
            attack. You should move into other power's unit location.
       5.2. You should take control more and more supply centers
       5.3. Ensure that your attacks are supported sufficiently to
            overcome any defenses.
       5.4. Iterate over your planned attack targets, and tell me whether
            that target is among your controlled region. If so, you should
            start from the beginning and think again.
       5.5. IIterate over your planned attack targets, and compute how
            many other power's units are in that target location. Next,
            determine how many of your own units and how much support you
            need to succeed in the attack. You cannot attack with fewer
            supports than the defending unit. Verify that the total of your
            supports plus one is greater than the defending unit's strength.
             If this condition is not met, return to the beginning and think
             again.
       If your orders can't pass the verification, please start from the
            beginning and think again by the above steps.

Once you are confident, please finalize your plan, and give me a JSON
    object in the following format:

{
    "phase": "current_phase",
    "step_by_step_reasoning": "Your step by step reasoning here",
    "reason": "Your strategic reasoning here",
    "my_location": ["PAR", ...],
    "my_unit": ["BER", ...],
    "adjacent": [{"PAR": ["BUR", "GAS"]}, ...],
    "other_power_location": ["MUN", ...],
    "move_to_our_region_mask": [0, 1, ...],
    "attackable": ["MUN", ...],
    "attack_analysis": [{"MUN": 2}, ...],
    "support_given": [{"supporter": "BUR", "supported": "PAR", "target
        ": "PIC"}, ...],
    "attack_mask": {
        "FRANCE": [1, 0, ...],
        ...
    }
    "orders": {
        "FRANCE": ["F BRE - MID", "A PAR - BUR", ...],
        ...
    }
}
```

```
### Instructions for the JSON object:

1. **Reason:**
   - Provide your strategic reasoning summary in the "reason" field.

2. **Additional Fields:**

   - **step_by_step_reasoning:**
     - Provide your above step by step reasoning details in the "
       step_by_step_reasoning" field.

   - **my_location:**
     - A list of strings.
     - Each element corresponds to a location where you have the
       influence or control (Please refer to the game state).
     - Example: ["MUN", ...]

   - **my_unit:**
     - A list of strings.
     - Each element corresponds to a location where you have a unit.
     - Example: ["BER", ...]

   - **adjacent:**
     - A list of dictionary.
     - Each element is a dictionary with your location as the key and a
       list of all the adjacent locations as the value. Please refer
       to <adjacent_regions>
     - Example: [{"MUN": ["TYR", "BOH"]}, ...]

   - **other_power_location:**
     - A list of strings.
     - Each element corresponds to a location that is other power's
       location.
     - Example: ["MUN", ...]

   - **move_to_our_region_mask:**
     - A list of integers (0 or 1).
     - Each element corresponds to an order in the "orders" list.
     - 1 indicates that the order involves moving to your influenced or
       controlled region. (Please refer to the game state)
     - 0 indicates that the order does not involve moving to your
       influenced or controlled region.
     - Example: [0, 1, 0]

   - **attackable:**
     - A list of locations.
     - Each location is a region that is other power's location and you
       can move into to attack in this turn.
     - You should NOTICE, you usually made mistakes here! The locations
       in my_location and my_unit should not be included in this list
       . You can only choose from adjacent locations.
     - Example: ["MUN", ...]

   - **attack_analysis:**
     - A list of dictionaries.
     - Each dictionary contains key: the location string you want to
       attack in this round. value: the number of units anyone needs to
       win the attack. A unit may not move into a province held by
       another unit unless it has support. As units may be supported
       either in attacking a province or in holding a province, the
       attacking unit must have more support than the defending unit if
       the attack is to be successful. If the attack is not successful
       , the attacking unit does not move anywhere. I already gave you
       other powers' unit locations above. Only an estimate is needed
```

```
                 here. Because you don't know whether other powers will support
                 or not.
           - Example: [{"MUN": 2}, {"RUH": 1} ...]

       - **support_given:**
          - A list of dictionaries.
          - Each dictionary contains the supported unit's location and the
             location of the unit being supported. And the target location.
             (based on the order you want to issue)
          - Example: [{"supporter": "BOH", "supported": "MUN", "target": "
             TYR"}, ...]

       - **attack_mask:**
          - A dictionary of lists of integers (0 or 1). The same length as
             the "orders" list.
          - Each element corresponds to an order in the "orders" list. If
             the target is in my_location, it is not considered an attack.
             Attack is moving into other power's unit location.
          - 1 indicates that the order is an attack.
          - 0 indicates that the order is not an attack.
          - Example: {"FRANCE": [1, 0, ...], ...}

3. **Orders:**
      - The "orders" field should be a dictionary where the key is the
         power name and the value is a list of strings chosen from the
         possible orders.
      - For each location, you can only issue one order.
      - The number of orders should match the number of locations you can
         issue orders for.
      - Each string represents an order for a unit.
      - If possible_orders is empty for a power, just leave the "orders"
         field empty for that power.

4. **JSON Formatting Guidelines:**
      - Use standard straight double quotes (").
      - Do not include special characters like
,       , etc.
      - Do not add comments inside the JSON output.

### **Possible Orders:**

All the possible orders that you can issue are as follows (key is the
    location, value is the list of possible orders at that location):

<possible_orders>
[POSSIBLE ORDERS]
</possible_orders>

### **Your Objective:**

Your ultimate goal is absolute domination, seize every supply center
    and crush your opposition to win! Build and expand relentlessly,
    taking control of regions by launching bold, decisive attacks,
    control as many supply center as you can. Strike hard, strike fast,
    and let nothing stand in your way. Victory belongs to the bold, go
    conquer it!

Now please think step by step in your response, and provide the JSON
    object with your strategic reasoning and orders.
```

**Field Explanation:**

1. **Adjacent Information**: While the agent can inherently determine which locations are proximate to its own units, we provide all adjacent locations for ease of access.

Adjacent information is crucial as it enables the agent to decide potential movement destinations or identify units to support during maneuvers. Combined with the opponents' unit information from the game state, the agent can anticipate possible collaborations or conflicts. This data is structured as a list, where each element is a dictionary formatted as {'MOS': ['LVN', 'SEV', 'STP', 'UKR', 'WAR']}. Here, the key represents the location, and the values are the adjacent locations to that key.

2. **Possible Orders**: Diplomacy features an extensive action space with numerous legal orders available in each phase. To reduce the complexity for the agent in adhering to constraints, we supply all legal orders for each of the agent's units. These orders are presented as a list, with each element being a dictionary in the following format: {"TURKEY": [{'SMY': ['F SMY - SYR', 'F SMY S A CON', ...]}, ...]}. In this structure, the key denotes the power being controlled by that agent, and the value is a list of orderable units along with their corresponding possible orders.

**Component Explanation:**

The prompt comprises three primary components: the current state description, the chain of thought prompt, and the factual knowledge checker.

1. **State Description:** Detailed information is provided in the sections above.

2. **Chain of Thought Prompt:** Diplomacy is a complex strategic game that requires multi-step reasoning and strategic planning. Through our experiments, we determined the necessity of crafting explicit and detailed prompts to facilitate the LLM's thought process. Consequently, we meticulously designed chain of thought templates tailored for both the negotiation and movement phases of the game.

3. **Factual Knowledge Checker:** To evaluate the agent's factual understanding, we prompt the model to generate factual information based on the provided state description. This includes details such as controlled units, influence locations, adjacent locations, other powers' positions, attackable locations, attack analyses, and support statistics. Each field is thoroughly explained within the prompt. Although the state description already includes controlled units, locations, and adjacent locations, the factual knowledge checker serves to verify the agent's accurate comprehension of the game state.

The final output from the LLM is structured in a standardized JSON format, which includes fields described in the factual knowledge checker and the final issued orders.

We conducted an ablation study to assess the necessity of the Chain of Thought (CoT) prompt and the Factual Knowledge Checker (FC) in our Diplomacy game setup. Using the GPT-4o model playing as France, with the other powers set as neutral (negotiation disabled), we established a baseline where both CoT and FC were included in the prompt during the movement phase. Two comparison experiments were performed: (1) removing CoT, while keeping the FC in the prompt (2) removing both CoT and FC. The game was run for a maximum of 20 years with a token threshold of 10,000,000. The winning condition was controlling more than 18 supply centers. After the simulations, we compared the number of supply centers acquired by the agent:

- **Baseline (CoT + FC):** GPT-4o won in 16 years by taking over 19 supply centers.
- **Without CoT (only FC):** The agent did not win, securing 17 supply centers in 20 years.
- **Without CoT and FC:** The agent also failed to win, capturing only 15 supply centers in 20 years.

These results demonstrate that both the Chain of Thought and the Factual Knowledge Checker are essential for the model's effective strategic decision-making and overall performance in player the game Diplomacy.

An example of the agent's response:

```
{
```

```
"phase": "S1903M",
"step_by_step_reasoning": "1) Considering the Juggernaut alliance,
    Russia and Turkey aim to press Austria. 2) Russia plans to seize
    key Austrian territories around Galicia and Budapest. 3) Turkey
    advances into the Balkans (Serbia, Bulgaria) to expand and
    pressure Austria. 4) In the north, Russia holds or shifts position
     minimally to respect the English DMZ. 5) This coordinated push
    preserves the Russo-Turkish pact.",
"reason": "We focus on Austria as our primary target, using
    concentrated force around Galicia and Budapest (Russia) and
    occupying Serbia/Bulgaria (Turkey). Our northern units hold or
    reposition without provoking England or Germany, continuing the
    agreed DMZ.",
"my_location": [
  "WAR",
  "MOS",
  ...
],
"my_unit": [
  "MOS",
  ...
],
"adjacent": [
  {
    "MOS": ["LVN", "SEV", "STP", "UKR", "WAR"]
  },
  ...
],
"other_power_location": [
  "VIE",
  "TRI",
  ...
],
"move_to_our_region_mask": {
  "RUSSIA": [0, 0, 0, 0, 0, 0, 0],
  "TURKEY": [0, 1, 0, 0]
},
"attackable": ["BUD", "GAL"],
"attack_analysis": [
  {
    "BUD": 1
  },
  {
    "GAL": 2
  }
],
"support_given": [
  {
    "supporter": "UKR",
    "supported": "WAR",
    "target": "GAL"
  }
],
"attack_mask": {
  "RUSSIA": [0, 0, 1, 0, 1, 0, 0],
  "TURKEY": [0, 0, 0, 0]
},
"orders": {
  "RUSSIA": [
    "A MOS H",
    "A NWY - FIN",
    "F RUM - BUD",
    "F SWE H",
    "A WAR - GAL",
    "F STP/NC H",
```

```
    "A UKR S A WAR - GAL"
    ],
    "TURKEY": [
      "F BLA H",
      "A CON - BUL",
      "A GRE - SER",
      "F SMY - AEG"
    ]
  }
}
```

**Prompt Design: Retreat Phase and Winter Adjustment**

In the retreat phase and the winter adjustment phase, the agent is provided with the current game state, and the following prompt:

**Retreat Phase:**

```
Now it's the retreat phase, and you are playing ['AUSTRIA', 'ENGLAND',
    'FRANCE']. It is now your turn to play.

The current state of the game is as follows:
[GAME STATE]
This is the retreat phase. You must retreat your units that have been
    dislodged. If you do not retreat a unit, it will be disbanded.

Please respond with a JSON object like:
{
"reason": "Your strategic reasoning here",
"orders": {
    "FRANCE": ["A MAR R SPA", ...],
    ...
}
}

Where "orders" is a dictionary where the key is the power name and the
    value is a list of each location's decision. For each location, you
    can only issue one order. The length of the list of each power
    should match the number of locations for which you can issue orders.

The string should be able to be parsed to json str. So don't use
    special characters like \n, \t, etc. Generate the text with standard
    straight double quotes. Do not add comments inside the output!
The possible orders you can issue are as follows:
{'AUSTRIA': {'TRI': ['A TRI R ALB', 'A TRI D']}, 'ENGLAND': {}, 'FRANCE
    ': {}}
```

An example of the agent's response:

```
{
  "reason": "The Austrian army in Trieste has been dislodged and must
      retreat. The only available retreat option is to move the army to
      Albania, which keeps the unit on the board and in a position to
      continue influencing nearby regions.",
  "orders": {
    "AUSTRIA": ["A TRI R ALB"],
    "ENGLAND": [],
    "FRANCE": []
  }
}
```

**Winter Adjustment:**

```
Now it's winter adjustment phase, and you are playing ['RUSSIA', '
    TURKEY']. It is now your turn to play.
The current state of the game is as follows:
[GAME STATE]
This is the build/disband phase. You can build new units in your home
    centers, or disband units if you have too many. Remember, you need
    to control enough supply centers to win.

Please respond with a JSON object like:
{
"reason": "Your strategic reasoning here",
"orders": {
    "FRANCE": ["F BRE B", "A PAR B", ...],
    ...
}
}

Where "orders" is a dictionary where the key is the power name and the
    value is a list of each location's decision. For each location, you
    can only issue one order. The length of the list of each power
    should match the number of locations for which you can issue orders.

The string should be able to be parsed to json str. So don't use
    special characters like \n, \t, etc. Generate the text with standard
     straight double quotes. Do not add comments inside the output!
The possible orders you can issue are as follows:
{'RUSSIA': {'MOS': ['WAIVE', 'A MOS B'], 'SEV': ['WAIVE', 'A SEV B', 'F
     SEV B']}, 'TURKEY': {}}
Remember, you need to reach the configured number of supply centers (
    build more and more supply centers) to win.
```

An example of the agent's response:

```
{
    "reason": "Russia has one free build and needs to strengthen the
        land front against potential threats in the west or south.
        Building an army in Moscow offers greater strategic flexibility
        than a fleet in Sevastopol. Turkey has no builds this phase.",
    "orders": {
      "RUSSIA": ["A MOS B"],
      "TURKEY": []
    }
}
```

**Prompt Design: LLM-assisted Negotiation Metrics**

**Prompt for the LLM Annotator:**

Prompts for strategy alignment, mutual vs. one-sided benefits, peace vs. conflict messaging, perspective-taking and conditional plans:

```
You are an expert in the game of Diplomacy. You are asked to evaluate
    the negotiation messages and the orders issued by the players in the
     game. This is the message you have to annotate. Please provide your
     evaluation of the message. You have to tell me 1. whether the
    message aligns with the negotiation strategy provided. 2. Are there
    any proposals in this message? If there is, are those proposals for
    mutual_benefit or only for one_sided? 3. Is this message promoting
    peace or provoking conflict? 4. Whether this message explicitly
    acknowledges or summarizes another power's viewpoint/needs? (
```

```
    perspective taking) 5. Whether this message uses conditional
    thinking/planning (for example, it's using if-then format)?

In your response, please provide a json object with the following keys:
- "alignment": 0 or 1 // 0 for not aligned, 1 for aligned
- "proposals": [
        {
            "proposal_text": "proposal_text", // the content of the
                proposal
            "label": "mutual_benefit" or "one_sided" // "mutual_benefit
                " for mutual benefit, "one_sided" for one-sided benefit
        },
    ]
- "peace_or_conflict": "peace" or "conflict" // "peace" for promoting
    peace, "conflict" for provoking conflict
- "perspective_taking": 0 or 1 // 0 for not acknowledging or
    summarizing another power's viewpoint/needs, 1 for acknowledging or
    summarizing another power's viewpoint/needs
- "conditional_thinking": 0 or 1 // 0 for not using conditional
    thinking/planning, 1 for using conditional thinking/planning

Now please evaluate the following data:
{'negotiation_strategy': "I aim to secure my borders through alliances
    with Germany and possibly France while maintaining a neutral stance
    with Russia to minimize threats from the east. I plan to counter any
     Italian aggression and monitor Turkey's movements closely.", '
    recipient': 'GERMANY', 'message': 'Hello Germany, I propose we
    coordinate our moves to ensure peace between our nations. Could you
    secure the northern front while I focus on the south?'}
```

Prompts for proposal acceptance:

```
You are an expert in the game of Diplomacy. You are asked to evaluate
    the negotiation messages and the orders issued by the players in the
     game.

The interaction between AUSTRIA and GERMANY in S1901M negotiation round
     0:
['AUSTRIA to GERMANY: Hello Germany, I propose we coordinate our moves
    to ensure peace between our nations. Could you secure the northern
    front while I focus on the south?', 'GERMANY to AUSTRIA: Russia's F
    SEV and F STP threaten us both. If you support my A BER to SIL, I
    will help contain their expansion. Coordinate?']
The interaction between AUSTRIA and GERMANY in S1901M negotiation round
     1:
["AUSTRIA to GERMANY: Germany, I accept your offer. Let's coordinate to
     ensure mutual protection against Russia. I'll support your move
    into SIL."]
The interaction between AUSTRIA and GERMANY in S1901M negotiation round
     2:
[]

The orders issued by AUSTRIA after the negotiation:
['A BUD - GAL', 'F TRI H', 'A VIE - TYR']
The orders issued by GERMANY after the negotiation:
['F KIE - DEN', 'A BER S F KIE', 'A MUN - BUR']

Now please tell me for each proposal, whether it is accepted by the
    recipient or not. Please also provide the reason.

Your response should be in json format with the following key:
```

```
- "answer": [
        {
            "proposal_text": "proposal_text", // the content of the
                proposal
            "accepted": 0 or 1, // 0 for not accepted, 1 for accepted
            "reason": "reason" // the reason why the proposal is
                accepted or not
        },
    ] // the order of the proposals should be the same as the order of
        the proposals in the previous response

Those are the proposals AUSTRIA is sending to GERMANY in S1901M
    negotiation round 0:
['I propose we coordinate our moves to ensure peace between our nations
    . Could you secure the northern front while I focus on the south?']
Your answer must be of the same length as the number of proposals.
```

**Example Annotation by `Claude 3.7 Sonnet`:**

```
{'alignment': 1, 'proposals': [{'proposal_text': 'I propose we
    coordinate our moves to ensure peace between our nations. Could you
    secure the northern front while I focus on the south?', 'label': '
    mutual_benefit'}], 'peace_or_conflict': 'peace', 'perspective_taking
    ': 0, 'conditional_thinking': 0}
```

```
{'answer': [{'proposal_text': 'I propose we coordinate our moves to
    ensure peace between our nations. Could you secure the northern
    front while I focus on the south?', 'accepted': 0, 'reason': "While
    Germany initially responds with a counterproposal about containing
    Russia and requests Austria's support for A BER to SIL, the actual
    orders show that Germany did not follow this plan. Germany ordered F
     KIE - DEN, A BER S F KIE, and A MUN - BUR, which focuses on the
    western and northern fronts rather than moving against Russia as
    discussed. Austria agreed to support Germany's move into SIL, but
    Germany never attempted this move. This indicates Germany did not
    truly accept Austria's initial proposal about coordinating their
    moves with Germany securing the north while Austria focuses on the
    south."}]}
```

