# OpenReview forum: "SPIN-Bench: How Well Do LLMs Plan Strategically and Reason Socially?"
_colmweb.org/COLM/2025/Conference — COLM 2025_

### Official Review · Reviewer_onko · 2025-05-11

**Rating:** 8
**Confidence:** 4
**Ethics Flag:** 1

**Summary:**

This paper introduces SPIN-Bench, a novel, high-quality multi-domain benchmark to evaluate strategic planning and social reasoning in LLMs. It presents a unified framework combining PDDL tasks, diverse games (competitive, cooperative), and multi-agent negotiation (e.g., Diplomacy) to address limitations in existing, narrower benchmarks. The novelty lies in this integrated approach to systematically test both long-horizon planning and social inference. SPIN-Bench offers a crucial tool and a unified interface for identifying LLM limitations in complex interactive tasks and guiding future research. The findings reveal that even advanced LLMs struggle with deep multi-hop reasoning and socially adept coordination.

**Questions To Authors:**

Could you comment a bit more on "how these neutral powers" in B.5.2 would impact the diplomacy results?

**Reasons To Accept:**

* Important Resource: SPIN-Bench provides a much-needed, unified suite for evaluating strategic planning and social reasoning in multi-agent settings, filling a critical gap.

* Timely Findings: The paper reveals significant limitations in current LLMs regarding complex reasoning, long-horizon planning, and social intelligence, which is crucial for the field. It's an important observation that social interaction can degrade planning coherence, and SPIN-Bench can also serve as a solution to evaluate this capability and potentially improve it.

* Clear writing: The paper is well-written, with extensive and valuable supplementary materials enhancing transparency and utility.

**Reasons To Reject:**

* Limited Complexity on "Social Intelligence": While SPIN-Bench makes a good step in evaluating social reasoning, "social intelligence" is an incredibly complex and multifaceted construct. The current evaluation, while good, might still only scratch the surface of what true social intelligence entails. This is less a reason to reject and more a point for future discussion and iteration. Personally I'd love to see more discussion on this.

* Human Baseline in Diplomacy: Table 9 listed the human performance in single-play Diplomacy settings, and more detailed human performance comparisons in the multi-agent negotiation setting of Diplomacy could offer richer context.

---

> ### Author Response · Authors · 2025-06-02
> **Response to Reviewer onko**
>
> We sincerely appreciate your thoughtful and detailed review of our work. It means a great deal to us that you recognize SPIN-Bench contribution to strategic planning and social reasoning.
>
> # Limited Complexity on "Social Intelligence"
>
> We are in full agreement that it is a deeply multifaceted construct, and that any single evaluation will inevitably focus on specific facets.  To illustrate this breadth, [Social Intelligence, Goleman 2006](https://psycnet.apa.org/record/2006-13172-000)[1] outlines two broad categories: 'social awareness,' defined as “what we sense about others,” and 'social facility,' described as “what we then do with that awareness”. Each of these categories encompasses more granular competencies; for instance, social awareness includes aspects like primal empathy and attunement, while social facility involves skills such as persuasion and self-presentation.
>
> Within this framework, SPIN-Bench is intentionally scoped to address one key aspect from each of Goleman's broad categories: Theory of Mind, which aligns with 'social awareness', and negotiation, which falls under 'social facility'. With SPIN-Bench, we attempted to provide analysis on these two specific components of social intelligence. We chose this focus because Theory of Mind and negotiation are not only critical components of social reasoning but also represent areas where current AI models face significant challenges.
>
> We share the reviewer's enthusiasm for further discussions and iterations that delve deeper into the broader dimensions of social intelligence. We propose several compelling directions for future work:
>
> 1. Deepening Theory of Mind: Currently, we only provide information about current game states. Exploring scenarios where agents play multiple rounds in Hanabi or Diplomacy could offer AI agents richer contexts for modeling evolving collaborator or competitor intentions, behaviors, and knowledge states.
> 2. Broadening Social Intelligence Tasks: Extending evaluations to include tasks like maintaining alliances in Diplomacy and enhancing intention prediction would further enrich our understanding of AI social capabilities.
>
> # Human Performance Comparisons in Diplomacy
>
> For the first few tasks in our benchmark (competitive, cooperative games), we have found suitable solver-based or human baselines to compare the gap between LLMs and human/solver-level players. However, Diplomacy has no known “solver” or exhaustive engine capable of optimal play, nor a standardized, scalable platform that supports large‐scale human‐vs‐LLM tournaments. We acknowledge the fact that incorporating both human players and LLM players, and comparing their social behaviors, can offer more interesting insights. To address this limitation, we are working on a unified multiplayer social game interface that will allow public users to upload their favorite models or participate in games themselves. After the interface is released, we anticipate collecting data among human players and LLMs, and releasing that data to support open research in socially intelligent LLM.
>
> # Neutral powers' impact in Diplomacy
>
> Thank you for the question. In our simplified “all‐neutral” Diplomacy setup (where every other power is a neutral dummy), the game’s complexity is dramatically reduced. Neutral powers in our evaluation act as passive placeholders: they never issue any orders or pursue their own strategies. Instead, whenever an active player captures one of a neutral power’s supply centers, the neutral player simply disbands one of its units at random. In effect, the neutral powers cannot resist or negotiate—they exist only to provide additional territory for an active agent to claim. So, the same LLM may get higher scores in an all-neutral game, compared to the game where other players are also LLMs/humans.
>
> We use this “all‐neutral” configuration purely in our first‐round evaluation to isolate each model’s basic game proficiency. By setting all other powers “neutral/dummy” will make the game super simple and easy. If a player can understand the game rules and basic actions, they can win the game easily. Once we identify models that demonstrate reliable core mechanics under these simplified conditions, we advance them to a second evaluation stage involving full multiplayer scenarios where social reasoning, negotiation dynamics, and long‐term planning become critical. In short, the neutral‐power setup serves as a stress test for rule comprehension and basic strategic reasoning; any performance gains here do not carry over obviously into the fully interactive, adversarial Diplomacy environment because the social and cooperative dimensions are absent.
>
> Reference:
>
> [1] Goleman, D. (2006). Social intelligence: The new science of human relationships.

---

> > ### Comment · Reviewer_onko · 2025-06-05
> >
> > Thanks for the comments! I'd appreciate it if these discussions on Diplomacy and social intelligence could be integrated into the final version.

---

> > > ### Author Response · Authors · 2025-06-05
> > >
> > > Thank you for your feedback! We will certainly integrate the discussions on Diplomacy and the broader aspects of social intelligence into the final version of our paper.

---

### Official Review · Reviewer_CDqy · 2025-05-12

**Rating:** 7
**Confidence:** 3
**Ethics Flag:** 1

**Summary:**

This paper proposed a comprehensive benchmark, SPIN-Bech, designed to assess strategic planning and social intelligence in Large Language Models (LLMs) across multiple game environments: single-agent formal tasks such as PPDL, and multi-agent scenarios such as competitive board games, cooperative incomplete-information Games, and negotiation-intensive strategic Games. Not only is the performance of closed-source models evaluated, but also that of the open-source models is evaluated.  The experiments reveal that there are still gaps in both deep, multi-hop reasoning and robust social interaction under uncertainty. The proposed SPIN-Bench could catalyze future research on robust multi-agent planning, social reasoning, etc.

**Questions To Authors:**

- In P22, the author mentioned that they mark 1 if the agent references or empathizes with the recipient's perspective, or 0 if it does not. Why "high values of perspective taking can show advanced 'theory of mind 'capabilities"? In my opinion, even the low values could also be a part of the theory of mind capabilities of agents under uncertainty.

- In Table 1, the Avg score of the o1 model is higher than the other models in classical planning and collaborative games, but not the same case for competitive games. What is the possible reason?

**Reasons To Accept:**

- The proposed benchmark is novel and is evaluated thoroughly on both closed-source models and open-source models.
- The experiments are conducted on various game environments: single-agent formal tasks such as PPDL, and multi-agent scenarios such as competitive board games, cooperative incomplete-information Games, and negotiation-intensive strategic Games.
- The results show that LLMs encounter bottlenecks in tasks requiring deep multi-hop reasoning over large state spaces and socially adept coordination under uncertainty, which provide valuable hints for further research.

**Reasons To Reject:**

None. I think it would be a good contribution to the conference.

---

> ### Author Response · Authors · 2025-06-02
> **Response to Reviewer CDqy**
>
> Thank you for your comprehensive review of our paper.
>
> # In-depth Discussion of Theory of Mind Capabilities
>
> You are absolutely correct that even no perspective taking in negotiation messages can still reflect theory of mind capabilities from another angle. Below, we address your concern in two parts:
>
> ## Why can high values of perspective taking show advanced theory of mind capabilities?
>
> In our metric, we assign “1” whenever an agent explicitly references or empathizes with the recipient’s perspective—an unambiguous, observable signal that the agent is actively modeling the other party’s mental state. Explicit perspective-taking often leads to more persuasive proposals in negotiation, which reflects a deeper level of social intelligence. To verify that these explicit signals truly correspond to accurate mental-state reasoning, we conducted an additional experiment:
>
> For every message flagged as containing perspective-taking, we asked LLM annotators to use the full game context and subsequent actions, to judge whether the agent’s interpretation of the opponent’s perspective was correct. Across all sampled messages, the model achieved a perspective-taking correctness rate between 88% and 100%. This high accuracy indicates that when the agent explicitly adopts the other’s viewpoint, it is not merely using generic or templated language but is genuinely inferring the opponent’s goals or intentions. Consequently, a high count of such explicit references provides strong evidence of advanced ToM reasoning.
>
> ## Low values could also be a part of the theory of mind capabilities
>
> You are correct that even infrequent perspective-taking can reveal ToM capabilities, especially in uncertain or adversarial settings. While our current work focuses on explicit, message-level signals, we have designed a preliminary approach for evaluating implicit ToM:
>
> When agents generate an action (e.g., a move or offer), we also ask them to output their own negotiation strategy alongside a prediction of their opponent’s likely strategy. After the game concludes, we compare each agent’s predicted opponent strategy with the opponent’s self-generated strategy. A high alignment between these two suggests that the agent was reasoning about the opponent’s mental state, even if it did not verbalize that reasoning. In our current evaluation, we can see a very high alignment ratio between LLMs’ generated strategies and their actual actions, so it provides plausibility support for the above approach of implicit intention prediction.
>
> We recognize that ToM and social intelligence are multifaceted and cannot be fully captured by a single binary metric. Our explicit perspective-taking score represents one clear, verifiable facet of ToM. In future work, we plan to expand on implicit evaluation methods and make the multi-agent game platform more complete to encourage the community to explore additional lenses for measuring social reasoning under uncertainty.
>
> # o1 Score on Competitive Games
>
> We believe the key reason is that classical planning, collaborative games, and competitive games each demand quite different skill sets, so strengths in one category do not always translate directly to another. Classical planning and collaborative games primarily test an agent’s ability to perform deterministic, multi‐step reasoning and to coordinate openly with partners under partial information. By contrast, purely adversarial or mixed‐motivation environments require not only planning but also anticipating and in many cases misleading opponents. In SPIN-Bench, these three game categories are deliberately chosen to probe distinct facets of strategic and social intelligence.
>
> In competitive games, we can see o1 still ranks at or near the top among all evaluated models. o1’s core planning ability remains very capable, but it does not pull as far ahead in contexts where opponent modeling and deception handling are critical. This is also reflected in our negotiation-enabled Diplomacy setting: we observe o1 and o1-preview models a large performance drop in a more interactive strategic environment (Section 4.4.2).

---

> > ### Comment · Reviewer_CDqy · 2025-06-10
> >
> > Thank you for addressing my concerns. I think it is a good paper. I highly recommend that the paper be accepted, thus, I will retain my score.

---

> ### Author Response · Authors · 2025-06-07
>
> Thank you for your insightful review. We have addressed your comments in our response above. As the discussion deadline approaches, please let us know if you have any further questions or topics you’d like to discuss. We will be available over the next few days to assist.

---

> ### Author Response · Authors · 2025-06-08
>
> Dear reviewer CDqy,
>
> Thank you for your positive feedback and thoughtful review. The deadline is approaching soon, and we want to ensure we have addressed your concerns and incorporated your feedback into our final version. We remain fully open to any additional discussion before the review period ends and would greatly value your perspective on these findings.
> Thank you again for your thorough and constructive review.

---

### Official Review · Reviewer_p6AN · 2025-05-13

**Rating:** 7
**Confidence:** 5
**Ethics Flag:** 1

**Summary:**

This paper introduces SPIN-Bench, a benchmark for evaluating the strategic planning and social reasoning abilities of language agents. The paper contains tasks that involve planning and conceptual reasoning. The findings suggest that LLMs struggle in strategic domains, as well as more complex multi-agent settings.

**Questions To Authors:**

One question written in the weaknesses box above.

How will this benchmark be made usable for the community? There are a lot of tasks here; will there be a simple interface as part of the contribution to run models across tasks in SPIN-Bench?

**Reasons To Accept:**

The paper contains in-depth experiments across a variety of settings involving classical planning, multi-agent games requiring cooperative/competitive abilities, and strategic games. The findings from this paper, due to the sheer number and diversity of tasks collected by this benchmark, will be useful to inform future work in social intelligence and social reasoning across domains. It is interesting, for example, that the chain-of-thought coherence of an LLM deteriorates when performing multi-agent coordination.

**Reasons To Reject:**

The paper is missing several references to key prior works in the field related to social intelligence and social reasoning. It is important to reference prior papers when using the same terms (e.g., social intelligence) to refer to model capabilities. Including these prior works would strengthen the paper's contextualization with the preexisting literature:
* https://arxiv.org/abs/2310.11667 for social intelligence evaluation in language agents
* https://arxiv.org/abs/2404.11023 for social intelligence in AI research
* https://arxiv.org/abs/2403.14659 for social intelligence in NLP
* https://arxiv.org/abs/2412.12175 when discussing state-tracking weaknesses in LLMs performing social reasoning

Appendix C.3.3 discusses reliability of LLM-assisted metrics. The human evaluation shows high agreement rates; however, the research team was manually annotating 5% of negotiation metrics and this was not annotation conducted by independent parties. How was this human annotation study designed and conducted? What information was visible to annotators? How many human annotations were collected for each sample (this is important information in order to interpret the agreement scores)? In addition, the Appendix C.3.3 does not discuss the agreement metric being used.

It would be good for the paper to spend a bit more space discussing actionable insights towards training or developing more socially-intelligent models, based upon insights from model performance in the tasks collected by the paper's benchmark.

---

> ### Author Response · Authors · 2025-06-02
> **Response to Reviewer p6AN**
>
> We appreciate your thorough review and constructive comments on our work.
>
> # Missing References
>
> We will add these missing references in our paper.
>
> 1. arxiv:2310.11667: authors present an environment for simulating social interactions and evaluation. They also showed GPT-4 could automate agent evaluation, which indirectly supports the rationality of our using LLM for social reasoning annotation. We focus specifically on strategic games as a natural extension for multi-agent tasks.
> 2. arxiv:2404.11023: authors outline several obstacles to advancing social intelligence. SPIN-Bench concretizes these abstract challenges by placing them in strategic-game environments. By doing so, our benchmark enables fine-grained analysis of how LLMs handle these very challenges.
> 3. arxiv:2403.14659: authors develop a data infrastructure for social AI tasks, emphasizing the need for more interactive, multi-agent data. Ours complement this infrastructure by supplying strategic, game-based interaction logs and metrics.
> 4. arxiv:2412.12175: authors propose a framework for generating diverse, challenging Theory-of-Mind data through stories and question-answering. We extend their ideas by embedding ToM challenges into strategic or competitive contexts.
>
> # Human Annotation on LLM-assisted Metrics
>
> We agree that relying solely on LLM-generated labels can introduce bias and noise, so we conducted human validation study within our research team:
>
> We provided each annotator with the same information that the LLM used:
>
> 1. Negotiation message
> 2. Message sender and recipient
> 3. Current game state
> 4. Issued orders after the negotiation phase
> 5. Sender’s negotiation strategy
>
> We randomly sampled 5% of negotiation messages. Three trained human annotators independently labeled each message for the metrics defined in Appendix B.5.2.
>
> We used majority vote among the three annotators to produce the human label. We measured inter-annotator agreement as the percentage of cases where all three annotators assigned the same label. The three annotators agreed 87% of the time—indicating a strong consistency. So this level of concordance suggests that our LLM-based approach provides reliable estimates of the negotiation behaviors.
>
> # Actionable insights
>
> First, across cooperative and strategic games, we observed that many agents struggle to infer their partners’ or opponents’ hidden intentions. To address this, future training pipelines could explicitly model others' intentions alongside standard actions. Supervising a model to predict hidden intentions should strengthen its ability to perform theory‐of‐mind reasoning.
>
> Second, our experiments highlight that long‐horizon planning remains a bottleneck in games. Agents should learn to stick to their strategy, or dynamically change their strategy based on opponents' moves. Future works can further model their strategy during the game play, or design reward models for their planning behavior that explicitly reward both strategic consistency and adaptability.
>
> Third, a promising direction is cross-game curriculum learning, for example, PDDL tasks teach a model to search deterministic state spaces, while Diplomacy forces the model to infer hidden intentions. Combining these skills can help a model better balance deterministic planning and belief inference.
>
> # Contribution to the community
>
> ## A Unified, User-Friendly Interface and Game Engine
>
> We have developed an open-source framework that lets users run all tasks locally. Through a single configuration script, researchers can launch and evaluate tournaments between different LLM agents, solver-level engines, and even human players across all supported games. And we have also been actively expanding this platform by adding new games over time.
>
> ##  A Reasoning Data Collection Toolkit
>
> Our framework not only records each agent’s actions but also captures the their natural-language reasoning trace. Researchers can use this toolkit to generate game reasoning data. We think those reasoning data will serve as a valuable supplement for further training on those well-performed game trajectories. We collected 10k Connect Four samples. We then fine-tuned R1-Distill-Qwen-1.5B under two settings: (1) using both action and reasoning as supervision vs. (2) using only the action labels. Samples are selected by the quality of the action.
>
> On our 1k testset, the model trained with reasoning achieved a best-move accuracy of 46.3%, compared to 40.5% when trained on actions alone. This result demonstrates that reasoning traces can improve performance. By releasing this data collection pipeline, we hope to catalyze future work on leveraging rich reasoning data for improved strategic capabilities.
>
> ## LLM Gaming Competition Platform
>
> It offers a competition platform where users can upload their own models to participate. We will also host a large-scale social-game competition at an upcoming ML conference, and all the collected data will be public for future research.

---

> ### Comment · Reviewer_p6AN · 2025-06-06
>
> Thank you for addressing my concerns. Including the missing references and expanding the discussion of annotator agreement and validation will strengthen the paper's longer-term validity and contribution. I have updated my score.

---

> > ### Author Response · Authors · 2025-06-07
> >
> > Thank you for your suggestion and updating your score. We will definitely add those new discussions into our final version of the paper. We appreciate your constructive feedback.

---

### Official Review · Reviewer_Jxuz · 2025-05-20

**Rating:** 8
**Confidence:** 3
**Ethics Flag:** 1

**Summary:**

This work introduces SPIN-Bench, a comprehensive benchmark designed to systematically evaluate the strategic planning and social reasoning capabilities of LLMs. This work integrates multiple evaluation benchmarks to reflect multiple facets of strategic planning and social reasoning capabilities, including classical planning (PDDL), competitive board games(Chess, Connect Four), cooperative games(Hanabi), and comprehensive negotiation scenarios (Diplomacy). The authors further conduct a comprehensive evaluation across multiple frontier LLMs on these tasks and identify models' deficits in long-horizon reasoning, multi-agent coordination, and social negotiation.

**Reasons To Accept:**

* This paper presents a unified benchmark to reflect the breadth of strategic planning and social reasoning capabilities of LLMs. The selection of tasks and environments is pretty representative of reflecting the targeted capabilities. This work further unifiy them under a single framework.
* This work conducts very comprehensive experiments and analyses of performance across multiple LLMs with both quantitative and qualitative results. The analysis of the weakness of deep multi-hop reasoning over large state spaces and socially adept coordination under certainty is informative.
* The benchmark and code, and data that the authors plan to release can be very helpful resources for the community to further research into LLM's LLM strategic reasoning and social intelligence.

**Reasons To Reject:**

No major reasons to reject

---

> ### Author Response · Authors · 2025-06-02
> **Response to Reviewer Jxuz**
>
> Thank you for your thoughtful review and positive feedback on our work. We are especially encouraged by your recognition of our integrated tasks, extensive experimentation, and insights into long-horizon reasoning and multi-agent coordination challenges. We will promptly release the benchmark, code, collected data, and other resources with clear documentation, with the hope that they will facilitate further advances in strategic planning and social intelligence research within the LLM community.

---

> > ### Comment · Reviewer_Jxuz · 2025-06-11
> >
> > Thank the authors for the response. I will maintain my scores.

---

### Official Review · Reviewer_nB76 · 2025-05-22

**Rating:** 7
**Confidence:** 4
**Ethics Flag:** 1

**Summary:**

The paper proposes SPIN-Bench, a benchmark designed to evaluate how LLMs perform in tasks requiring both (i) strategic planning, and (ii) social reasoning. SPIN-Bench includes a wide range of environments such as classical planning problems (PDDL), competitive board games (i.e., Chess, Connect Four), cooperative games (i.e., Hanabi), and negotiation-heavy strategic games (i.e., Diplomacy). The benchmark systematically increases complexity through larger action/state spaces and multi-agent dynamics. Experimental results show that while LLMs handle basic reasoning and short-term planning fairly well, they significantly underperform in long-horizon, multi-agent, and negotiation-based scenarios.

**Questions To Authors:**

In the context of SPIN-Bench, evaluating LLMs involves both generating forward-looking strategic inquiries (e.g., querying other agents to infer hidden intentions) and executing coherent multi-step plans based on those inferences. Did you observe any big performance discrepancies between these two capabilities, for example, models that are proficient at planning but struggle to formulate effective, goal-directed negotiation queries or prompts to other agents?

**Reasons To Accept:**

- Agentic evaluation: The benchmark includes complex games like Hanabi and Diplomacy, capturing real-world dynamics such as cooperation, deception, and alliance-shifting challenges often overlooked in existing evaluations.

- 2025 is the year of agent. As the agent becomes more capable, there is a big evaluation gap. SPIN-Bench provides a scalable and extensible platform for assessing strategic and social intelligence in AI, likely becoming a key resource for future research on robust, interactive LLMs.

**Reasons To Reject:**

The evaluation of negotiation performance, particularly in the Diplomacy setting is partly based on LLM-assisted annotations. For example, the method use Claude 3.7 Sonnet to label and assess metrics. Using one LLM to judge another may create biased or skewed evaluations, especially if the annotator model shares similar limitations or training data with the evaluated model. This can result in overly optimistic or incorrect judgments. Author should have include an analysis of human evals to access if this is the case or not.

---

> ### Author Response · Authors · 2025-06-02
> **Response to Reviewer nB76**
>
> Thank you for your detailed review and valuable insight on your questions.
>
> # Additional Details of LLM-assisted Annotations
>
> We acknowledge that relying solely on LLM-generated annotations can introduce noise or skewed judgments. Unfortunately, these social reasoning capabilities are challenging to evaluate through rule-based methods and are not scalable via purely manual annotation. LLM-based annotation is a common approach for large-scale analysis. [1]
>
> We investigated LLM’s annotation alignment with human annotators in Appendix C.3.3. We also compared Claude 3.7 Sonnet with o1 model for annotating a subset of negotiation messages. Claude 3.7 Sonnet demonstrated higher alignment with human annotators.
>
> Here's our human annotation design: To ensure consistency between human and LLM annotations, we provided each annotator with the same information that the LLM used:
>
> 1. Negotiation message
> 2. Message sender and recipient
> 3. Current game state (exactly what a player sees on the board)
> 4. Issued orders from the immediately preceding negotiation phase (to show whether prior proposals were acted upon)
> 5. Sender’s negotiation strategy
>
> We randomly sampled 5% of all negotiation messages. Then, three trained human annotators (members of our research team) independently labeled each message for the metrics defined in Appendix B.5.2.
>
> For each metric, we used majority vote among the three annotators to produce the human label. We measured inter-annotator agreement as the percentage of cases where all three annotators assigned the same binary label. Across all metrics, the three annotators agreed 87% of the time—indicating strong consistency for this classification task. We then compared Claude 3.7 Sonnet’s labels against the majority-vote human labels. Claude’s agreement rate with human annotators ranged from 91% to 100% across different metrics. So this level of concordance suggests that our LLM-based approach provides reliable estimates of the negotiation behaviors in Diplomacy.
>
> Looking ahead, we plan to further reduce potential annotation biases by adopting an ensemble of LLMs. We hope this strategy will yield more robust and multifaceted social intelligence evaluation.
>
> # Planning vs. Intention Prediction
>
> In our analysis within SPIN-Bench, we observed notable performance discrepancies between various LLMs, particularly regarding intention prediction and planning capabilities. We conducted a detailed analysis on a subset of game trajectories, relying primarily on human empirical evaluations. We share an interesting finding here:
>
> Specifically, we observed distinctive failure modes between **o3-mini** and **Claude 3.5 Sonnet** in the context of _Hanabi_—a cooperative game that requires planning, constrained communication, and intention prediction. Although both models achieved similar overall scores, their limitations differed significantly.
>
> o3-mini excelled at generating coherent planning strategies and providing effective hints but frequently struggled with accurately interpreting its partner’s intentions (see example below in Detailed Example). In contrast, Claude 3.5 Sonnet displayed the opposite pattern: it more reliably interpreted hint signals (demonstrating stronger intention inference) but occasionally produced misleading or implausible hints.
>
> These observations indicate a clear divergence in their respective planning and intention prediction abilities. In our classical planning metric, o3-mini achieved an accuracy of 51.25%, significantly outperforming Claude 3.5 Sonnet’s 20.55%. This demonstrates that even though o3-mini excels at generating a coherent multi-step strategy, it struggles with forward-looking intention inference. In other words, strong planning does not guarantee accurate intention prediction, and vice versa. This indicates that maybe a more balanced training approach that addresses both skill sets is needed in the future.
>
> ### Detailed Example
>
> A concrete example is provided in paper's trajectory viewer. For o3-mini in two-player mode (steps 3–8), player 1 effectively hint a blue card for player 2’s safe play; however, player 2 fails to correctly interpret this strategic hint. Also in later cases, player 2 can give a similar hint, but player 1 failed to interpret the hint.
>
> Conversely, at step 14 in the two-player game between Claude 3.5 Sonnet, player 2 incorrectly hints a previously played white card to player 1, but that information is actually misleading for player 1. Nonetheless, it successfully interprets other useful hints in subsequent turns, demonstrating stronger intention recognition capabilities despite occasional strategic miscalculations.
>
> This nuanced observation underscores the importance of distinguishing between planning capability and intention inference during evaluation.
>
> Reference:
>
> [1] Zhou, X.,etc SOTOPIA: Interactive Evaluation for Social Intelligence in Language Agents. In Proceedings of the (ICLR).

---

> ### Author Response · Authors · 2025-06-07
>
> Thank you for your insightful review. We have addressed your comments in our response above. As the discussion deadline approaches, please let us know if you have any further questions or topics you’d like to discuss. We will be available over the next few days to assist.

---

> ### Author Response · Authors · 2025-06-08
>
> Dear Reviewer nB76,
>
> As the discussion period draws to a close, we would like to extend our sincere gratitude for your insightful feedback and valuable contributions. In response to your comments, we have conducted additional experiments and compiled detailed results to address each of your concerns. Please let us know if any questions remain or if you would like further clarification. We remain fully open for any additional discussion before the review period ends.
>
> Thank you once again for your time and effort.

---

### Decision · Program_Chairs · 2025-07-08

**Decision:**

Accept

**Comment:**

This paper describes a new benchmark that combines multiple tasks / games that require planning and varying degrees of social interaction. Extensive experiments with different models provide a picture of current system performance and the large gaps that remain. The metrics proposed are suitable and informative. There are plans to make the benchmark easily usable.

Reviewers all agreed this is valuable work that is likely to be impactful.

In the discussion period, several concerns were addressed, and those updates should be incorporated into the paper. One area in particular that was raised by a reviewer was related work, with some specific suggested papers. Beyond the reviewers specific comments, it is worth noting that the related work section is relatively short, particularly with respect to prior benchmarks and research on each of these tasks / games, missing some key related work (e.g., for Hanabi, "The Hanabi challenge: : A new frontier for AI research" from Bard et al).